# Context-dependent selectivity to natural images in the retina

**Matías A. Goldin** [1,5] ✉, **Baptiste Lefebvre**[1,2,5], **Samuele Virgili** [1,5], **Mathieu Kim Pham Van Cang**[1,3], **Alexander Ecker** [4], **Thierry Mora**[2], **Ulisse Ferrari** [1] & **Olivier Marre** [1] ✉

Retina ganglion cells extract specific features from natural scenes and send this information to the brain. In particular, they respond to local light increase (ON responses), and/or decrease (OFF). However, it is unclear if this ON-OFF selectivity, characterized with synthetic stimuli, is maintained under natural scene stimulation. Here we recorded ganglion cell responses to natural images slightly perturbed by random noise patterns to determine their selectivity during natural stimulation. The ON-OFF selectivity strongly depended on the specific image. A single ganglion cell can signal luminance increase for one image, and luminance decrease for another. Modeling and experiments showed that this resulted from the non-linear combination of different retinal pathways. Despite the versatility of the ON-OFF selectivity, a systematic analysis demonstrated that contrast was reliably encoded in these responses. Our perturbative approach uncovered the selectivity of retinal ganglion cells to more complex features than initially thought.

To carry out complex visual tasks, it has been proposed that each area of the visual system extracts different features from the visual scene. The complexity of these extracted features increases along the hierarchy of visual areas[1]. This feature extraction process starts in the retina, whose output is composed of retinal ganglion cells. These cells can be classified in different cell types[2] and each of them is supposed to extract a low-level visual feature from the visual scene[3]. This feature selectivity is often inferred by reverse correlation using a white noise stimulus[4]. This type of linear response modeling using stimuli with simple statistics, although effective in determining an approximate receptive field of the cell, falls short of modeling the full nonlinear response profile of the retina to complex stimuli such as the ones it is usually exposed in nature.

A basic example of feature extraction is the selectivity of ganglion cells to luminance increase or decrease: the so called ON cells are those sensitive to light increments, whereas OFF cells are those sensitive to light decrements, and ON-OFF cells respond to both. Recent works have shown that, surprisingly, this ON-OFF selectivity depends on the visual context. For example, it changes with the background luminance[5,6]. Transient variations of the preferred polarity have also been reported in response to large motion in the periphery[7]. In these studies, retinal processing was probed with simple artificial stimuli. It is not clear how ON-OFF selectivity would generalize to more complex stimuli, even in a regime of constant average global contrast and luminance[8].

Here we use a novel perturbative approach for probing context-dependent selectivity with perturbations added on top of natural scenes. We stimulated ganglion cells of mouse and axolotl retinas with natural images, and then added small checkerboard-like perturbations on top of them. These perturbations evoked small changes on the responses of retinal ganglion cells. When analyzing systematically the responses to these perturbations, we found that the same ganglion cell can be selective to light increments when the perturbations are added on top of one natural image, and to light decrements when they are added on top of another. Ganglion cells can thus switch their selectivity from ON to OFF depending on the context, and do so during natural scene stimulation. We designed a non-linear model to explain

[1]Institut de la Vision, Sorbonne Université, INSERM, CNRS, Paris, France. [2]Laboratoire de physique de l'Ecole normale supérieure, CNRS, PSL University, Sorbonne University, and University of Paris, Paris, France. [3]Institut de l'Audition, Institut Pasteur, INSERM, Paris, France. [4]Institute of Computer Science and Campus Institute Data Science, University of Göttingen, Göttingen, Germany. [5]These authors contributed equally: Matías A. Goldin, Baptiste Lefebvre, Samuele Virgili. ✉e-mail: matias.goldin@inserm.fr; olivier.marre@inserm.fr

and predict these changes, and mapped this model to specific circuits in the retinal network. Finally, we demonstrated that this strong context dependence is compatible with a robust computation of a more abstract feature: contrast.

## Results

### A new method to estimate selectivity during natural stimulation

We recorded ganglion cells in the retina of mice and axolotls with multi-electrode arrays (MEAs) while stimulating photoreceptors with flashed natural images. Each stimulus was presented for 300 ms, followed by a gray screen of 300 ms. To measure the selectivity of ganglion cells during natural image stimulation, we added dim checkerboard patterns (Fig. 1a) to natural images. The amplitude of the checkerboard pattern was chosen so to elicit a small but visible change on the average ganglion cell response compared to the response to the same unperturbed natural image (see Supplementary Fig. 1). We repeated many times the presentation of the same "reference" natural image, but we perturbed it each time with a different checkerboard pattern. To avoid

any adaptation to the reference natural image, we interleaved these stimuli with other perturbed natural images, and with a large number of flashed natural images without any perturbation. All natural images had the same average luminance and contrast (see Methods).

For each cell and each reference image, we estimated a *local spike-triggered average* (named hereafter LSTA) (Fig. 1b), as the average of the perturbation patterns weighted by the number of spikes they evoked. This estimation is similar to a classical Spike Triggered Average (STA)[4], but here the checkerboard patterns have a small amplitude and thus explore a small, local region of the stimulus space centered on the reference natural image. A large number of cells showed a detectable ON or OFF LSTA (48 out of 50 cells from 2 axolotl retinas; 443 out of 634 cells from 6 mice retinas, see Methods).

### Ganglion cells can change their selectivity to luminance and space in different natural contexts

For many cells (around 86% for mice, 83% for axolotl), all the LSTAs were consistent with the classical STA estimated with a standard

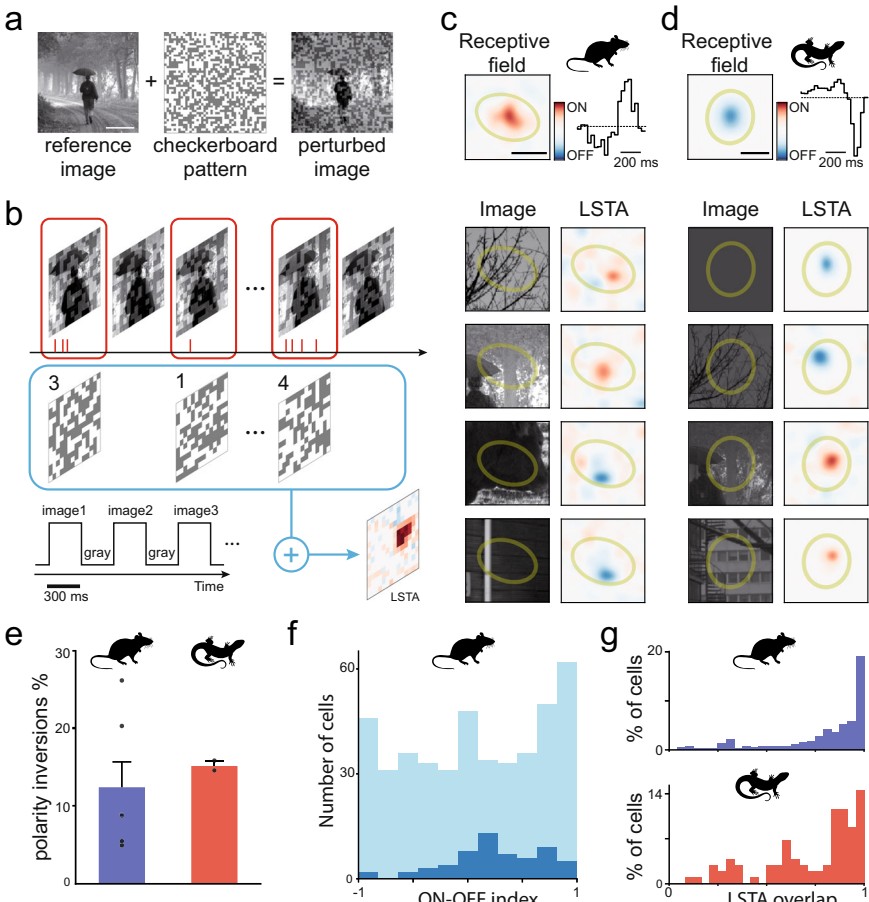

**Fig. 1 | ON-OFF selectivity can change in a natural context for the same ganglion cell. a** We added a random checkerboard pattern with low contrast on top of a natural image to obtain a perturbed image. Scale bar: 500 μm. **b** We flashed a randomized sequence of perturbed and unperturbed images. All flashes lasted 300 ms and were separated by 300 ms of gray whose luminance was equal to the mean luminance of the images. To calculate the LSTA, we averaged the different perturbations, weighted by the number of spikes they evoked. **c, d** Two examples of LSTAs, for mouse (ON cell, **c**) and axolotl (OFF cell, **d**), measured for two example cells and different perturbed images. Top: the classical receptive field with its spatial (left) and temporal (right) components. A green ellipse fitted to the spatial component is shown in all the bottom panels as a reference for each cell. Bottom left: reference image; bottom right: corresponding LSTA. The first two rows LSTAs match the classical polarity, and the last two rows show a polarity inversion. Scale bars: 200 μm. **e** Percentage of ganglion cells showing both ON

and OFF polarities for different natural images (*N* = 6 for mouse and *N* = 2 for axolotl). Data are presented as mean and SEM for mouse and as mean and semi dispersion for axolotl. **f** Number of ganglion cells recorded in total (light blue) and showing LSTA polarity inversion (dark blue) against their ON-OFF index estimated from their responses to full-field flashes. −1 means a pure OFF cell, 1 a pure ON, and 0 an equal response to both ON and OFF flashes. **g** Cell population distribution of the overlap value between pairs of LSTAs for the same cell, for mouse (top) and axolotl (bottom). Overlap is estimated as the absolute value of the denoised normalized scalar product between two LSTAs from two different natural images (see Methods). 1 means that the two LSTAs had the same position, 0 that they had non-overlapping positions. *N* = 2527 overlaps were calculated for mouse and *N* = 103 for axolotl. Source data are provided as a Source Data file. Credit for the natural images shown here goes to Hans Van Hateren: http://bethgelab.org/datasets/vanhateren/.

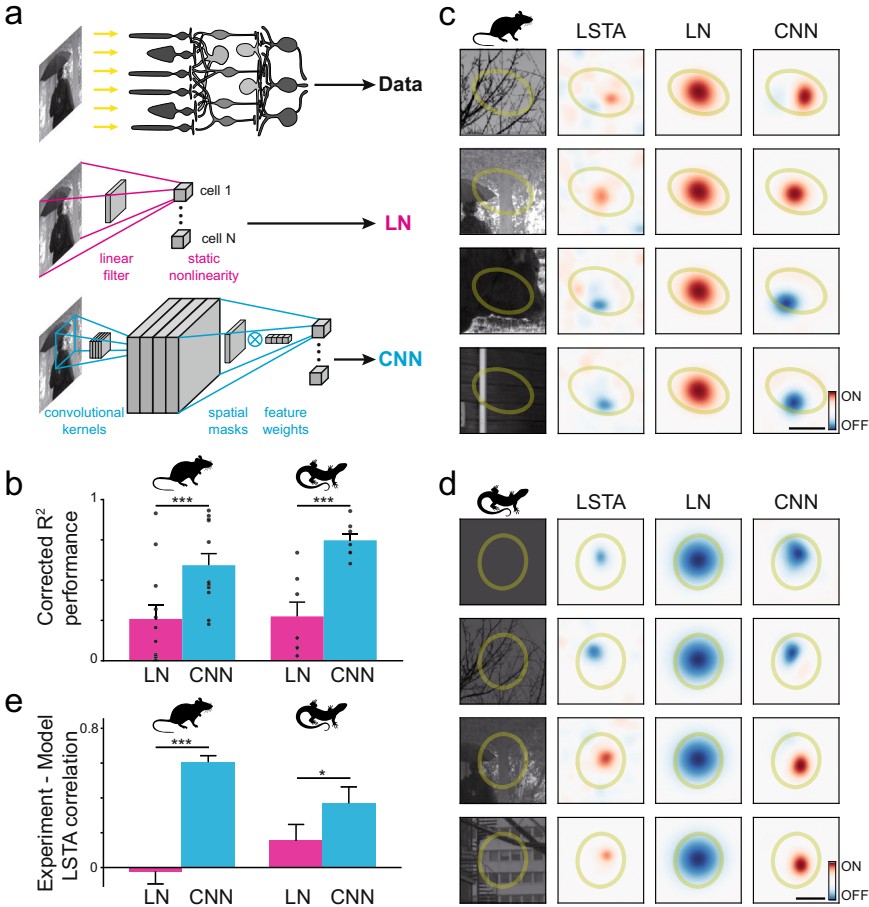

**Fig. 2 | A non-linear, convolutional Neural Network model predicts the shape of the LSTAs. a** Schematic of the different architectures used to predict the responses of multiple retinal ganglion cells to a flashed image (top). The LN model (middle) is composed by a linear filter followed by a nonlinear function. The CNN model (bottom) is composed of a convolutional layer (inferred kernels) and a dense layer (readout weights factorized in spatial masks and feature weights). **b** Average performance of the two models at predicting the average responses to repeated, unperturbed natural images (see Methods and Supplementary Fig. 4 for a scatter plot of this data across all modelled cells). The data reported here is from $N = 12$ (mouse) and $N = 7$ (axolotl) cells that were both modelled and showed polarity inversion. In both species the CNN significantly outperforms the LN ($p = 1 \times 10^{-3}$ for mouse and $p = 1 \times 10^{-2}$ for axolotl, two-sided Wilcoxon signed-rank test). **c**, **d** Two example LSTAs (second column), for mouse (**c**) and axolotl (**d**), measured for two example cells (same as Fig. 1) and different reference images (first column), along with the prediction of the two models (third and fourth column for LN and CNN models, respectively). **e** Average performance of the two models at predicting the LSTAs of inverting cells (see Methods). The data reported here is from $N = 57$ (mouse) and $N = 26$ (axolotl) measured LSTAs. Again in both species the CNN significantly outperforms the LN ($p = 1 \times 10^{-9}$ for mouse and $p = 5 \times 10^{-2}$ for axolotl, two-sided Wilcoxon signed-rank test). Bar plots are presented as mean and SEM. Source data are provided as a Source Data file. Credit for the natural images shown here goes to Hans Van Hateren: http://bethgelab.org/datasets/vanhateren/.

checkerboard (Fig. 1c, d, top). However, we also found many cases where a single ganglion cell shows an ON-type LSTA for one natural image and an OFF-type for another one (Fig. 1c, d, bottom). This means that, when stimulated with one image, the neuron may increase its firing rate if the luminance is increased inside its receptive field. However, when stimulated with a different image, the firing rate may increase if the luminance inside the receptive field decreases. This result shows that the ON-OFF selectivity (termed hereafter polarity) is not a fixed feature of this ganglion cell, but varies depending on the context.

A significant fraction of ganglion cells showed this inversion in the polarity of the LSTA (Fig. 1e; $14 \pm 4\%$ for mouse and $16.7 \pm 0.7\%$ for axolotl). This result did not seem restricted to a single cell type. 39% of ganglion cells showing polarity inversion could be classified as "ON-OFF" (Fig. 1f, ON-OFF index between −0.25 and 0.25, see Methods). All of them had a clear dominant ON or OFF component in their classical receptive field measured with standard checkerboard. A large fraction of ganglion cells showing polarity inversion is thus receiving inputs from both ON and OFF pathways, but the natural context seems to be able to modulate their respective contributions. Ganglion cells with

polarity inversion were also more often transient than sustained cells (Supplementary Fig. 2).

We looked further into the cell classification using two stimuli, a chirp[2] and drifting gratings[9] (see Methods and Supplementary Fig. 3). We found that the inverting cells in these experiments do not belong to a specific type, but can appear in several groups. We found that all cells showing polarity inversion had at least a detectable response to both ON and OFF flashes. Cell types with a "pure" ON or OFF selectivity did not show polarity inversion. However, the ratio between the ON and the OFF responses could vary very broadly: some cells (e.g. the ON transient type) had a strong ON response and a barely detectable OFF response to flashes, but showed polarity inversion for some cells when probed with natural images (see Supplementary Fig. 3, and Supplementary Table 1). Only one inverting cell was direction selective.

Beyond the changes in polarity, we observed displacements of the LSTA from one reference image to another. To characterize these displacements, we measured the amount of overlap between the different LSTAs of the same cell by estimating the absolute value of a normalized scalar product between all the pairs of denoised LSTAs from the same cell (see Methods). This measure will give 1 if the two

LSTAs fully overlap, and 0 if they are disjoint. We found that overlap was low for a fraction of them (Fig. 1g). These results show that both spatial and luminance selectivity are dependent on the natural context.

## A convolutional neural network model can account for context dependence

Is there a model that could explain this context dependence? We tested if different types of models could explain our results. To do this we showed the retina, besides the perturbed images described above, an independent and interleaved set of unperturbed natural images. We fitted quasilinear and nonlinear regression models to predict the activity of the ganglion cells from the stimulus and then we used the fitted models to predict the LSTAs (see Methods). The quasilinear regression model was a conventional linear-nonlinear model[4,10]. It filters the visual stimulus linearly, and then uses a monotonic nonlinear function to transform the result of this filtering into an output neural response (Fig. 2a, middle).

For the non-linear model, we used a nonlinear encoding model implemented using convolutional neural networks[11]. In this type of network, a set of higher-level representations (feature maps) are extracted from the visual stimulus and combined to predict neural responses (see Methods). Our network was composed of two layers: a feature extraction layer followed by a readout layer. Each layer was composed of a linear filter followed by a nonlinearity. The first layer was a convolutional layer with four two-dimensional convolutional kernels, where the filters were learned from the data. For each experiment, the four kernels were common to all the modelled cells. In the second layer there was one filter followed by the same non-linearity for each ganglion cell (Fig. 2a, bottom, see Methods)[12,13]. In what follows, we refer to this model as the convolutional neural network model (CNN). Note that none of the models were trained with the perturbed natural images. Only unperturbed flashed natural images were used for training.

To estimate the model performance, we first tried to predict the average response to another set of flashed natural images that were not used for training, and were repeated multiple times. For each cell we estimated an R-square value corrected to take into account the limits imposed by the noise (see Methods) on a cell by cell basis. The CNN model outperformed the LN model both in mouse ($60 \pm 3\%$ vs $43 \pm 3\%$, $p < 10^{-3}$, two-sided Wilcoxon signed-rank test) and axolotl ($73 \pm 3\%$ vs $48 \pm 4\%$, $p < 10^{-2}$) (Fig. 2b). Note that the images used for performance estimation were not part of the training set. Nonlinear models are thus necessary to make accurate predictions on a stimulus ensemble composed of unperturbed natural images.

We next investigated whether these models could generalize, and could also reproduce the LSTAs obtained for different natural images, and in particular the changes in polarity described above. To predict the LSTAs with a model, we calculated the gradient of the model output with respect to its input for each ganglion cell and each natural image (see Methods). Note that the reference images for which the experimental LSTAs were measured were not part of the training set. Figure 2c, d shows the comparison between the shape of the LSTAs observed experimentally and the ones inferred by each model, for the same cells as in Fig. 1. As expected, the LN model always predicted the same LSTA shape, independent of the reference natural image. The LN model could thus not reproduce the observed changes in LSTA polarity or location. On the contrary, the LSTAs predicted by the CNN model were very similar to the experimental ones. We quantified the quality of this prediction by estimating the correlation between model-predicted LSTAs and experimental LSTAs after denoising (see Methods) for the cells that showed polarity changes. The average correlation was significantly better for the CNN model than for the LN model (Mann-Whitney U test, $p < 10^{-8}$ for mouse and $p < 0.05$ for axolotl; Fig. 2e). These results show that the CNN model was able to predict both the neural responses to flashed natural images and the

corresponding LSTAs, even if the model was never trained with these perturbed images, much better than the LN model. The CNN model was thus able to generalize and not only predict the responses to images, but also to perturbations of them.

An interpretation of these results is that the LSTA describes the linear function that would best approximate the stimulus-response function locally. The LSTA corresponds to the perturbation that would increase the most the firing rate of the cell, if applied to the corresponding natural image. LSTA could thus be seen as an experimental estimate of the local gradient of this function[14]. Our results regarding the polarity inversion for different images can thus be interpreted as large changes in the gradient of the stimulus-response function over the stimulus space. Simple linear models have a constant gradient and cannot explain these results. Linear models followed by a monotonically increasing non-linearity cannot explain polarity inversion either, as the non-linearity only scales the gradient by a non-negative number. A special case is a model composed of a linear filter followed by a quadratic non linearity[15]: since the non-linearity is not monotonic, it multiplies the gradient by a number that can be either positive or negative. This model can thus show polarity inversion. However, it predicts that the gradient varies only by a scaling factor, with no change in shape. As a result, it cannot predict the observed displacement of the LSTAs (Fig. 1c, d, g). Models with more non-linearities, like the CNN model, are thus necessary to accurately predict the observed changes in both spatial and luminance selectivity.

## Polarity changes are due to ON and OFF pathway convergence onto ganglion cells

Which component of the CNN model allows predicting the LSTA changes, and which could be the corresponding pathways in the retinal circuit? The first layer of the CNN is composed of convolutional kernels. After learning on ganglion cell responses, they have either a strong ON center and a weaker OFF surround, or an OFF center combined with a weaker ON surround (Fig. 3a and Supplementary Fig. 5A, B). This is analogous with the processing performed by ON and OFF bipolar cells. The second layer pools the rectified outputs of this first layer, and is analogous to the way ganglion cells sample inputs from bipolar cells. In ganglion cells where LSTA polarity inversion was observed, the learned model pooled inputs from both ON and OFF kernels of the first layer (Fig. 3b). If we removed the ON kernels from the model, polarity inversion would disappear. This suggests that retinal ganglion cells show LSTA polarity changes because they pool inputs from both ON and OFF bipolar cells, directly or indirectly, and that inactivating the ON bipolar cells could suppress the observed polarity inversion. We tried clustering the modeled cells in the 4D feature weight space, to see whether the inverting cells would cluster together. However, we found no clear patterns in the feature space (Supplementary Fig. 5C). This might be due to the small number of cells involved per experiment which prevents performing robust clustering.

To test if LSTA changes of polarity depend on the ON pathway, we estimated LSTAs while blocking the ON pathway in the retinal circuit (Fig. 3c). We first measured LSTAs for different reference images, then blocked the transmission from photoreceptors to ON bipolar cells in the mouse retina by adding L-AP4 to the bath (see Methods), and repeated the same stimulus to measure LSTAs again. We detected across 3 mice retinas a total of 209 cells showing LSTAs before L-AP4 application. Of these, 26 were detected to invert. In retinal ganglion cells presenting LSTA polarity inversion, LSTAs with ON polarity disappeared (Fig. 3d, top two rows), while LSTAs with OFF polarity were largely unchanged (Fig. 3d, bottom two rows). In only in 4% of the cases, LSTAs with ON polarity remained after L-AP4 application, while LSTAs with OFF polarity did stay in 26% of the cases (Fig. 3e). The decrease of LSTAs with OFF polarity is due to the experiment duration when adding L-AP4 and measuring again LSTAs: ganglion cells tend to

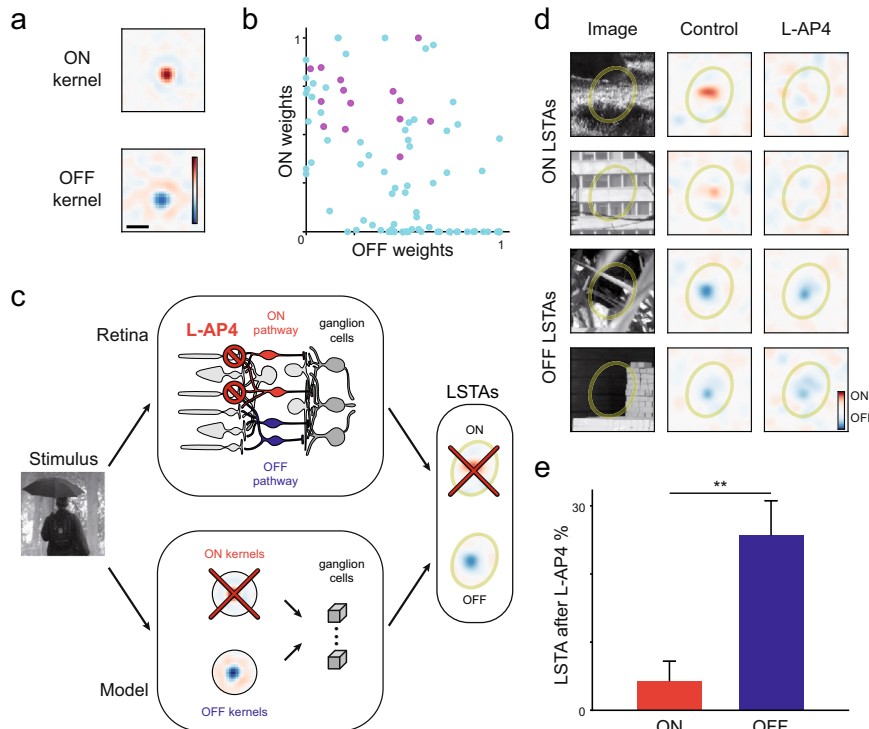

**Fig. 3 | ON-OFF pathway convergence onto ganglion cells is needed for LSTA polarity inversion. a** Examples of learned convolutional kernels (scale bar: 200 µm) showing both ON and OFF polarities. All kernels present a strong center and a weaker surround of the opposing polarity. The kernels shown here are taken from mice, as all the rest of the data in the figure (see Supplementary Fig. 5A, B for more details). **b** Feature weights from ON (y-axis) and OFF(x-axis) feature maps for each modeled ganglion cell. Purple: ganglion cell with LSTA polarity inversion. **c** Schematic showing the model-retina analogy. A stimulus arrives at the first processing stage of the CNN, the convolutional kernels, which are analogous to the bipolar cells in the middle layer of the retina. Information from the kernels is pooled by the modeled ganglion cells, or combined into ganglion cells in the output layer of the retina through the ON and OFF pathways. Both modeled and biological ganglion cells can show ON and OFF LSTAs. For the analogy with the pharmacology experiment, the ON kernels of the model are suppressed (red cross), which

corresponds to the application of L-AP4 blocking synaptic transmission between photoreceptors and ON bipolar cells. In both the model and the biological retina, ganglion cells cannot show ON LSTAs, while OFF LSTAs remain unchanged. **d** Example cell from mouse in a pharmacology experiment. This cell is a classical ON cell, which displays both ON and OFF LSTAs (Control column) for different images. After applying L-AP4 (see Methods), ON LSTAs disappear (top two rows) while OFF LSTAs remain unchanged (bottom two rows). **e** Proportion of LSTA measured in control and still present after bath application of L-AP4, across all cells displaying polarity inversion. The difference is highly significant ($p = 2 \times 10^{-3}$, two-sided Fisher exact test). The data was obtained on $N = 26$ inverting cells across 3 experiments. The errors for the two percentages are estimated from a binomial distribution. Source data are provided as a Source Data file. Credit for the natural images shown here goes to Hans Van Hateren: http://bethgelab.org/datasets/vanhateren/.

---

lose sensitivity after many hours of recording, and this results in some LSTAs disappearing over time. Nevertheless, the difference in decrease between the LSTAs showing ON and OFF polarity is highly significant ($p = 2 \times 10^{-3}$, Fisher exact test). As a result, ganglion cells that showed an inversion of polarity in the control condition did not after L-AP4 application. Our pharmacological manipulation thus suppressed the observed inversion of LSTA polarity as predicted (Fig. 3e). The changes of LSTA polarity thus require the convergence of the ON and OFF pathways onto the same ganglion cell.

**Changes in polarity underlie a robust encoding of local contrast**
We showed above that the CNN model can accurately predict the LSTAs of retinal ganglion cells for different natural images. To investigate more systematically the relationship between the reference natural image and the corresponding LSTA, we took advantage of the learned CNN model to predict and generate LSTAs corresponding to each of the 2910 reference natural images used for training the model, for each modeled ganglion cell (see Methods). For each of these cells we performed a principal component analysis (PCA) on the ensemble of predicted LSTAs. For most cells, the first two principal components (Fig. 4a) accounted for a large fraction of the variance of these LSTAs ($87 \pm 3\%$ across population). We thus projected all the LSTAs, as well as the natural images themselves, in the two-dimensional space formed by these two first principal components. We represented each image

as a point in this two-dimensional space, and the associated LSTA as a vector (Fig. 4b, c). We chose this vector field representation because the LSTA represents the optimal direction in the stimulus space in which to perturb the stimulus in order to increase the firing rate of the ganglion cell (i.e. the gradient of the stimulus-response function). For readability, we binned the projection space and averaged together all the images and the corresponding LSTAs that fell in the same bin.

For some ganglion cells this representation shows that the direction of the LSTAs is almost always pointing in the same direction (Fig. 4d). This corresponds to a cell with little variation in the LSTA shape and polarity across images. In this case, an increase of firing rate will always signal a change in the same direction in the stimulus space, no matter what the reference image is. This direction corresponds here to a decrease of the local luminance (same as PCA 1 in Fig. 4a).

On the contrary, other ganglion cells showed large variations in the direction of the LSTAs in the PCA space (Fig. 4e, f). This corresponds to important changes in the LSTAs across images, like the ones present in cells with inversion of polarity. However, these changes were not random: there was a systematic relation between the reference image (i.e. the starting point of the arrow) and the LSTAs (i.e., the direction of the arrow). For these cells, the vector field showed a diverging structure: the vector size was minimal at a central point and LSTAs always pointed outward from this central point (Fig. 4e, f black arrows). The vector field representation thus uncovered a systematic

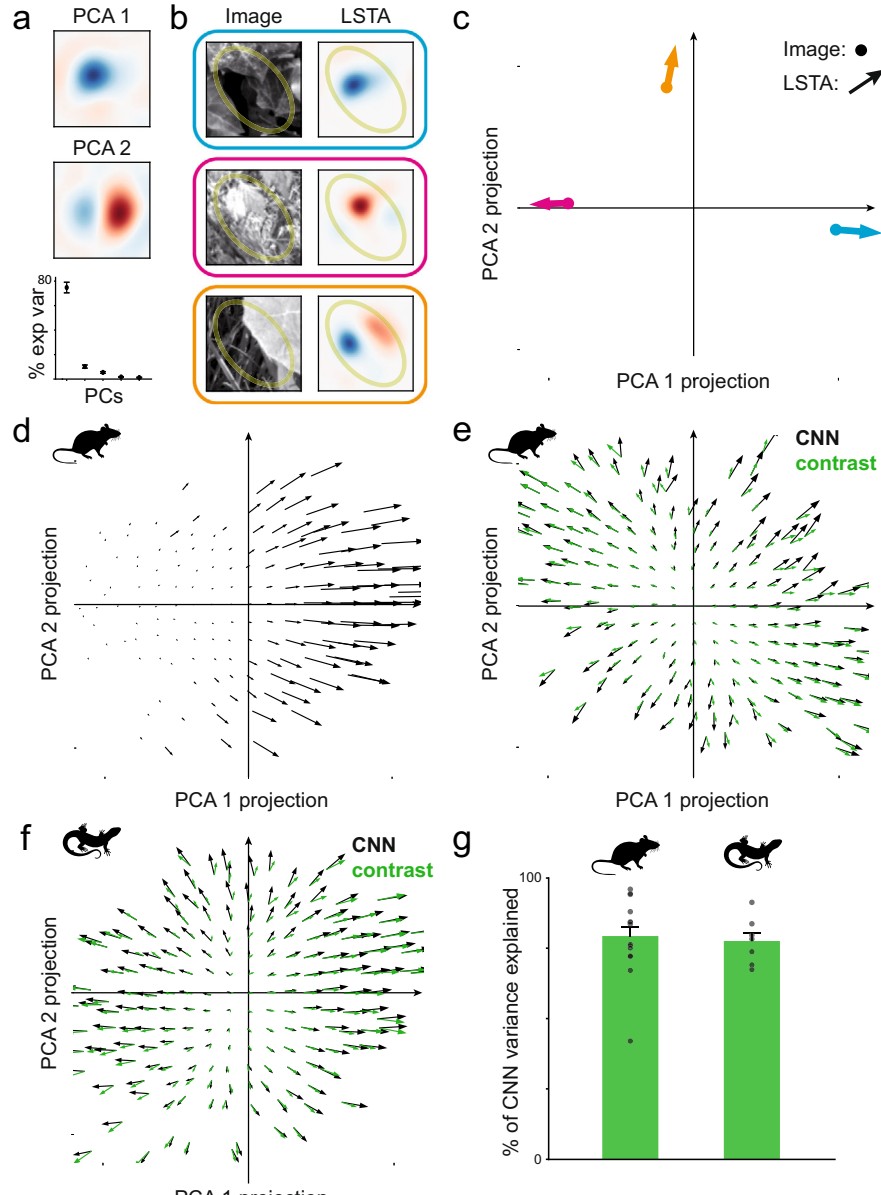

**Fig. 4 | There is a systematic relationship between images and LSTAs. a** For each cell, we predicted the LSTAs associated with 2910 natural images with the CNN model. We applied PCA on this ensemble of LSTAs for each cell. Top and middle row: the first two PCA components for an example mouse retina ganglion cell. Bottom row: variance of the LSTA ensemble explained by the first five principal components, averaged across all the inverting cells of mouse. **b**, **c** Representation of image-LSTA pairs as points and arrows in the two dimensional PCA space representation for the stimulus space. In (**b**) three image-LSTA pairs are color coded into point-arrow pairs. In (**c**) the arrow origins are located on the projection of the image in the PCA space and the arrow indicates the projection of the LSTA. **d** Vector field for an example mouse cell encoding for luminance. The resulting vector field points in the direction of the first principal component (PCA 1, see **a**), showing that this cell increases its firing rate for decreasing luminance inside the

receptive field, regardless of the image chosen as context. **e** Vector field for another ganglion cell. In this case, the resulting vector field (black arrows) diverges from the center. This means that this cell will fire more for a bright image if it gets brighter inside the receptive field, and for a dark image if it gets darker inside the receptive field. Green arrows: prediction of the contrast model (see text). **f** Same as (**e**) for an example axolotl cell. **g** Performance of the contrast model at predicting the variance in the neural response predicted by the CNN (see Supplementary Fig. 6 for a scatter plot of this data). The data reported here is from $N = 14$ (mouse) and $N = 7$ (axolotl) cells that were both modelled and showed polarity inversion. Data are presented as mean and SEM. Source data are provided as a Source Data file. Credit for the natural images shown here goes to Hans Van Hateren: http://bethgelab.org/datasets/vanhateren/.

---

relation between the reference images and their corresponding LSTA shapes.

Since the LSTA corresponds to a gradient, this central point corresponds to a local minimum of the stimulus-response function. The single minimum and the structure of the vector field with a positive divergence is reminiscent of the one produced by a quadratic function. Therefore, we hypothesized that this vector field could be produced by a cell whose response is proportional to the

square difference between the presented image and a uniform gray image, which is a contrast function[16]. We thus defined a local contrast function as the square difference between the luminance value and a constant gray value, taken for each pixel and averaged over the receptive field center. We generated the vector field predicted by this local contrast function (see Methods, and Fig. 4e, f green arrows). The two vector fields, the one generated with the CNN model and the one generated with this local contrast function, looked similar.

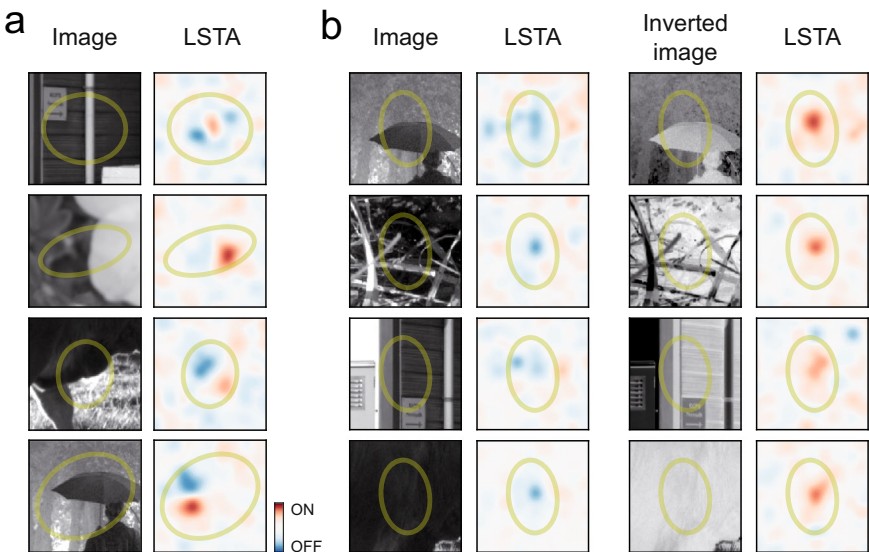

**Fig. 5 | Contrast computation is confirmed using natural images and their inverted bright-dark counterparts. a** Four example images presenting high contrast in the center of the receptive field of four mouse ganglion cells. As hypothesized (see text), bright (dark) regions in the image correspond to bright (dark) LSTA regions. **b** An example of OFF ganglion cell of mouse retina where LSTA was measured both for natural images (1st column) and for the negative images (3rd column). LSTAs that appeared dark for the images, were bright for the negative counterparts. Credit for the natural images shown here goes to Hans Van Hateren: http://bethgelab.org/datasets/vanhateren/.

Despite having more parameters, the CNN model was only slightly better at predicting the responses to natural images ($58 \pm 7$ % vs $48 \pm 6$% for mouse; $74 \pm 4$% vs $46 \pm 6$% for axolotl). The local contrast model could explain on average 79% of the variance of the responses observed in the CNN model for mouse and 78% for axolotl (Fig. 4g), and largely outperformed the LN model ($p$-value=$5x10^{-4}$ for mouse and 0.03 for axolotl). This suggests that the computation performed by the CNN model on those cells can be reasonably approximated by a computation of local contrast.

This vector field representation uncovers a systematic relation between LSTA and the corresponding natural image, and this relation suggests that these ganglion cells can compute local contrast when stimulated with natural images. To check if this relation could be found directly in our data, without relying on the CNN model, we first looked for several examples where the stimulus falling inside the receptive field center was a high contrast one, with both dark and bright regions. If the responses of a ganglion cell are increased by an increase of contrast inside the receptive field, any perturbation that would make a bright region brighter, or a dark region darker, should increase the firing rate. As a consequence, the measured LSTA should have an OFF region in the darkest regions of the reference image, and a ON region in the bright region of the natural image. We found several examples where there was a striking correspondence between these dark and bright regions of the natural image and the corresponding LSTA (Fig. 5a).

To further confirm this correspondence, we performed experiments where we presented reference natural images, and the corresponding negative images, where black becomes white and white becomes black. We measured the LSTAs for each of these images. If bright regions in the reference image become dark regions, and vice-versa, following our interpretation, the corresponding LSTA should also change and take the opposite polarity. Among the cells detected to invert (39 out of 527 cells recorded in 4 mouse retinas), $57 \pm 9$% were observed to reliably change their LSTA polarity when presented with negative natural images (Fig. 5b). These results confirm that ganglion cells that show polarity inversion in their LSTAs can perform a robust local contrast computation, despite the changes in their ON-OFF selectivity.

## Discussion

We have shown that, during natural image stimulation, many ganglion cells can change their selectivity for light increase or decrease depending on the natural image. This strong dependence of selectivity on the context can be modelled by non-linear neural network models, and a convolutional neural network model learned on the data can reproduce these results. These changes in the ON-OFF selectivity result from a convergence of the ON and OFF pathways onto the same ganglion cells. The changes in polarity of the LSTA suggest strong context-dependent responses. However, our results show that this is compatible with the robust encoding of more abstract features like contrast.

We could only estimate LSTAs for a limited number of reference images. It is thus possible that some ganglion cells that did not show change in LSTA polarities in our results would have shown a polarity inversion with a different choice of reference natural images. Therefore, the exact ratio of cells showing this effect might be underestimated. On the ganglion cells where the CNN model was learned, we predicted LSTAs for 2910 natural images, and found that a larger fraction of these cells should present changes in LSTA polarity according to the CNN model (54% in mouse, 66% in axolotl). Note that it is also possible that this polarity inversion could also be observed for stimuli other than natural images. For example, we observed it for negative images (Fig. 5).

Our results show that contrast can be robustly decoded from these ganglion cells by reading their spike count, since their response always increases with local contrast, no matter what the reference image is. On the contrary, information about local luminance is ambiguous. For some reference natural image, increasing in firing rate is associated with an increase of local luminance, and with a decrease for other images. However, the information about luminance might be preserved through other decoding methods. For some cells we noticed that the latency for the response to natural images associated with an ON-like LSTA was different from the one associated with an OFF LSTA (Supplementary Fig. 7). It might thus be possible to read luminance from the latency of the response. Previous works have suggested that information about different features of the visual scene might be encoded in different features in the retinal response, e.g., latency and

firing rate[17,18]. Our results support this idea in the case of natural stimulation. In order to account for the different response latencies, we implemented a simple extended version of our CNN model to make it time dependent (see Supplementary Fig. 8). Although this simple modification predicted the response latency well for some cells, it made important mistakes for others. Improving the performance further would require to try very different model architectures (e.g., Vierock et al.[19] McIntosh et al.[20]) and to learn the model on responses to natural movies, rather than flashed natural images.

Previous works have shown that OFF ganglion cells can respond to light onset when changing the background luminance[5]. Here we have measured the ON-OFF selectivity of ganglion cells for natural images where the overall background luminance was kept constant. Furthermore, in our stimuli all the image presentations, with and without perturbations, were interleaved to avoid any adaptation to the specific statistics of a natural image. The changes in polarity cannot thus be attributed to adaptation to the background luminance.

Geffen et al.[7] reported that OFF ganglion cells can switch to ON polarity during and slightly after a sudden shift of a large grating present in the surround. Compared to our study, the change in that study was evoked by a transient and dynamical change in one stimulus property, while we measured the LSTAs for flashed natural images. The results of Geffen et al.[7] suggest that extending our paradigm to a spatio-temporal case[14,21], by applying perturbations on top of a scene with natural dynamics, could reveal even more complexity in the LSTA dependence on context.

Maheswaranathan et al.[22] learned a deep network model on retinal responses to natural movies and noticed that the gradient of the stimulus-response function, estimated with their network model, could change polarity depending on the content of the natural scene. The gradient they calculated is the closest equivalent to our LSTA. The main difference with our approach is that they only estimated it on a model, while our estimation of the LSTA allows a direct experimental measure of how retinal processing can be context-dependent during natural image processing. Our method to measure LSTA can be used to validate models by showing a qualitative difference between different model classes. For example, quasilinear models could not predict polarity changes or displacements. In comparison, classical estimation of performance cannot tell what is missed in the prediction of the response.

Measuring LSTAs has thus two interests. First, it is useful to test possible models of sensory processing, and possibly discard some. The changes observed in the measured LSTAs could simply not be reproduced by linear-nonlinear models. Second, it can also be a tool to test which features of the image can be robustly coded by a ganglion cell. Our results showed that ganglion cells presenting changes in LSTA polarity could perform a robust computation of local contrast.

Related works in the visual cortex have learned deep neural networks online to then find the stimulus that maximized the neuronal response[13,23–25]. This approach is complementary to ours: while they looked for the most exciting input, LSTAs can be seen as the locally most exciting input, i.e. the local change that will be best at increasing the response. An interesting outcome of our study is that these local changes do not always point in the same direction. Instead, this direction can systematically vary with the reference image. Our results thus show that the processing performed by one neuron on the visual scene cannot be summarized by a single most exciting stimulus, and calls for a deeper evaluation of single neuron selectivity and invariance[26] when stimulating with natural scenes by using LSTA estimation.

Similarly, a related work by Keshishian et al.[27] in the auditory cortex, using deep network models learned on data, approximated the processing performed by neurons in the auditory cortex by describing three different STAs for one cell, and each of them was associated with a different region of the stimulus space. In our vector field representation, this would correspond to three regions inside which the vector field would be constant. Similarly to Maheswaranathan et al.[22], that study did not measure them experimentally. More importantly, our results show that even in the retina some ganglion cells cannot be reduced to a small number of STAs because they code for a more abstract feature, which results in systematic changes in the LSTA depending on the reference image. Our perturbative approach could thus be applied in other sensory systems to refine models or test hypotheses about what features are extracted from the sensory input.

## Methods

### Electrophysiological recordings
Electrophysiological data were recorded from isolated retinas from 6 C57BL6J mice of 8 to 11 weeks (median of 9), and 2 adult axolotl salamanders (Ambystoma mexicanum, pigmented wild-type). The animals were housed in enriched cages with ad libitum food, and watering. The ambient temperature was between 22 and 25 °C, the humidity was between 50 and 70% and the light cycle was 12–14 h of light, 10–12 h of darkness. The experiment was performed in accordance with institutional animal care standards of Sorbonne Université. After killing the animal, the eye was enucleated and transferred rapidly into oxygenated Ames medium. Dissection was made under dim light condition as described previously[28,29]. We mounted a piece of retina onto a membrane, and then lowered it with the ganglion cell side against a 252-channel multi-electrode array (MEA) whose electrodes were spaced by 30 μm. During dissection and recordings, the tissue was perfused with oxygenated Ames solution and a peristaltic perfusion system with 2 independent pumps: PPS2 (Multichannel Systems GmbH). Mice retinas were kept at 35–37 degrees and axolotl retinas at room temperature (20–24 degrees) during the whole experiment.

The data sampling rate was 20 kHz. The raw signal was acquired through MC_Rack Multi-channel Systems software 4.6.2, it was high-pass filtered at 100 Hz, and the spikes were isolated using SpyKING CIRCUS 1.0.6[28]. Subsequent data analysis was done with custom-made Python codes. We extracted the activity of a total of 634 neurons from mice, and 50 neurons from axolotl. We kept cells with a low number of refractory period violations (< 0.5%, with median < 0.09% for all experiments, 2 ms refractory period) and whose template waveform could be well distinguished from the template waveforms of other cells. These constraints ensured a good quality of the reconstructed spike trains. In addition, we discarded neurons that showed no or almost no responses to flashed images, preventing the estimation of LSTAs.

### Pharmacology
For the experiments requiring the ON-pathway inactivation, we used a metabotropic glutamate receptor agonist (L-AP4, Tocris Bioscience) to block synaptic transmission between the photoreceptors and the ON-bipolar cells. A new Ames medium was preheated at 35–37 degrees with L-AP4 diluted at 5 μM concentration and used to perfuse the retina. To evaluate the effectiveness of L-AP4, we stimulated the retina with full-field flashes at 0.25 Hz and checked that the spiking responses to the onset of the flash were abolished.

### Visual stimulation
A white mounted LED (MCWHLP1, Thorlabs Inc.) was used as a light source, and the stimuli were displayed using a Digital Mirror Device (DLP9500, Texas Instruments) and focused on the photoreceptors using standard optics and an inverted microscope (Nikon). The light level corresponded to photopic vision: $4.9 \times 10^4$ and $1.4 \times 10^5$ isomerisations / (photoreceptor. s) for S cones and M cones respectively.

### Checkerboard stimuli
We displayed a random binary checkerboard during 40 min to 1 h at 30 Hz to map the receptive fields of ganglion cells. Check size was

42 µm for mice and 73.5 µm for axolotl. For mice retinas, a second checkerboard stimulation was shown before the end of the experiment to control for the stability of the ganglion cell functional responses.

A three dimensional STA (x, y and time) was sampled using 21 time samples. The spatial STA presented across all the figures was obtained as the 2 dimensional spatial slice at the maximum value after smoothing. The temporal STA is the one dimensional time slice at that same value. A double Gaussian fit was performed on the resulting spatial STA, and the ellipse corresponding to a 2σ contour of the fit was plotted for all the figures.

### Natural image stimuli

We used the Open Access van Hateren Natural Image Dataset[30], which consists of 4212 monochromatic and calibrated images taken in various natural environments. The calibration ensures a strictly linear relationship between scene luminance and pixel value. To avoid that the retinal system adapts to different ranges of light intensities encountered in different environments, we performed a preprocessing step. First, we identified the images with a significant number of pixels above saturation, which was defined as the proportion of pixels above a given threshold (6266 for ISO 200, 12551 for ISO 400 and 25102 for ISO 800). If the saturation level was above 2%, the image was discarded. This resulted in a total of 3190 images. Second, we cropped the central part of each image (final size: 864 × 864 px). Third, for each image, pixel values were converted to luminance with the conversion factor provided by the calibration. Fourth, the images were normalized with a custom procedure to fix the mean luminance and the root mean square (RMS) contrast: the linear scale was transformed to log scale, the distribution of pixel values centered and scaled, to finally come back to the linear scale and centered and scaled the distribution of pixel values for a second time to a final mean and standard deviation (respectively 0.5 and 0.25 for mice, and 0.36 and 0.12 for axolotl). Pixel values below 0 were clipped to 0, and those above 1 to 1.

### Unperturbed natural image stimulus

The dataset of 3190 images obtained after the preprocessing step described in the previous paragraph was used to stimulate the retina. We selected 30 images which were presented multiple times (30 and 20 repetitions each for mice and axolotl respectively) to create the test dataset for the CNN. The other images were only flashed once. Around 10% of them (250 images both for mouse and axolotl) were allocated to the validation dataset while the rest of the images composed the training dataset.

### Perturbed natural image stimulus

The generation of the perturbed natural image stimulus consisted in superimposing some of the natural images (7–8 for mouse and 4 for axolotl) with multiple perturbation patterns. We used a checker size of 42 µm for mice and 54 µm for axolotl, and the checkerboard had 72 × 72 checks for mice and 56 × 56 checks for axolotls.

### Calibration of perturbation amplitude

We determined the minimal perturbation amplitudes that produced a discernible spike count difference between the response to the natural image and to the corresponding perturbed natural image. In a calibration experiment, we used 4 images and several fixed perturbation patterns for mice and axolotl respectively, that were repeated 25 times for each of 7 perturbation amplitudes, and were presented in a randomized manner. The final perturbation amplitude chosen for mice was 12.5% and 3.125% for axolotl, where a value of 100% corresponds to the maximum intensity, used for a classical checkerboard. These values corresponded to a change in the firing rate of approximately 1.5 Hz, in the ganglion cells that responded with a high enough firing rate to the unperturbed images (Supplementary Fig. 1).

### LSTA calculation

For each reference image, we counted the spikes occurring between 30 and 350 ms after each perturbed image presentation. We then did an average of the perturbation patterns of each reference image weighted by their corresponding spike counts. For plotting, we applied a smoothing by bicubic splines. An ellipse corresponding to the 2σ contour of the classical STA was plotted on top as a reference in all figures.

To count the number of ON and OFF LSTAs, we denoised the LSTA using a spline-based method for receptive field estimation[31], with 2 additional steps to only detect LSTA that were distinguishable from background and compact in space.

To measure the overlap between LSTAs of the same cell (displacement, Fig. 1g), the weighted average results were denoised by a gaussian blur and then were fitted with a 2D gaussian envelope. The overlap between each couple of LSTAs was then estimated as the normalized scalar product of their fitted gaussians, in absolute value.

To estimate the quality of the model-based LSTA predictions (Fig. 2e), the LSTAs of the modelled cells were again denoised as described above with a gaussian blur and fitted with a 2D gaussian. Their fit was then correlated pixelwise with the model-based LSTA predictions.

### ON-OFF index

We presented 20 repetitions of full field illumination steps to the retinas. An ON step of 3 s duration was followed by an OFF step of the same length. To measure the ON-OFF index of ganglion cells, we defined a window of 800 ms after each step onset. The ON-OFF index was calculated as the difference between the sum of spikes occurring in the window after the ON step and the sum of spikes after the OFF step, divided by the total spike count in both windows.

### Cell typing

We performed three further experiments in mice retinas of C57BL6J mice of 17 weeks to find out if our polarity inverting cells do belong to any specific cell type.

**Stimulus.** In addition to our perturbation protocol to detect polarity inverting cells, we applied two additional ones. 1) A full field 'chirp' stimulus composed of ON and OFF steps, plus varying full field frequencies and amplitudes, with luminance values ranging from 0 to 1. The stimulus is the exact same that Baden et al.[2] used to find and classify 32 different types of ganglion cells. It was played at 50 Hz, containing 20 repetitions of 32 s length, (See Supplementary Fig. 3). 2) Drifting gratings (DG) moving in 8 different directions with a speed of 479.5 um/s, at a spatial period of 959 um (274 pixels at 3.5 um/pix) and at 50% Michelson contrast (0.75–0.25 luminance). Each DG lasts 10 s, preceded by 2 s of gray (0.5 luminance), the temporal period being 2 s. Therefore, each grating edge goes through a ganglion cell's receptive fields 5 times per DG. The 8 directions were repeated 4 times in a pseudo-random manner. The stimulus profile and dynamics is identical to the one described in Yao et al.[9] to retrieve direction selective cells. In our case we used a unique luminance value, as described in the Visual stimulation section above.

**Typing.** To cluster cells in different types, we based our analysis on the chirp and checkerboard stimulus responses, and representing each ganglion cell with a reduced representative vector. To obtain these vectors, first we constructed peri-stimulus histograms (PSTH) from the spikes evoked from the chirp stimulus, using a binning of 100 ms. Then, for each experiment, we z-scored all PSTHs and performed a PCA on them. We kept the number of components that were needed to explain 80% variance of the data (around 12 components). Second, we used the temporal profile (21 samples at 30 Hz) of each cell's STA obtained using the checkerboard stimulus. We z-scored it and

performed a PCA, keeping the first component, which explains around 60% of the variance. This adds information about the classical STA polarity of the ganglion cells. Third, we used the area of the ellipse fitted to the classical STA, as the product value of their major and minor axis σ values. These areas were normalized from 0 to 1. In this way, we obtain a data vector of around 14 values, depending on the experiment, that describes each ganglion cell according to their response to a chirp and a checkerboard. Then, we performed an agglomerative clustering, setting the threshold value in a way that all clusters look homogeneous across PSTHs and STAs. This resulted in overclustering that produced around 50 ganglion cell groups (from around 200 cells in each experiment). In the last step, we assigned each cluster group to one of the 32 types described in Baden et al.[2]. To do this, we used the Calcium imaging data provided by the authors to match it with our data. We based ourselves in their Extended data Fig. 1, where the authors link electrophysiology and calcium imaging by means of a convolution between a $Ca^{2+}$ event triggered by a single spike. We transformed our PSTHs by convolving them with a decaying exponential, in which we adjusted the temporal decay constant to maximize correlation of our cluster groups and theirs (median maximum correlation of 0.76). Cell types that present strong responses to the modulating frequencies and amplitude were assigned correctly, while other types which mostly respond to ON/OFF steps, were assigned in a second round of correlation match after excluding the former groups. Besides the correlation of the chirp traces, we confirmed the correct assigning of groups by checking that the ellipses of each type form a proper mosaic, that the spatial STAs look uniform, the similarity of their direction selectivity PSTHs (see below) and that of the spike waveforms. Finally, we computed a correlation matrix between the average chirp response of each type to show that the groups are homogeneous (Supplementary Fig. 3A).

**Direction selectivity.** We constructed PSTHs from the spikes evoked from the chirp stimulus, and calculated the mean firing rate evoked by each DG direction, and normalized it to the maximum direction for each cell (values 0 to 1). To assess selectivity, we calculated the vector sum of these normalized response vectors, which spanned values from 0 to 2, as it is usually done[2,9]. To test if a neuron was direction selective, we performed a shuffle test on the cells whose resulting vector sum was bigger than 0.5, randomly permuting the direction labeling of every DG trial, and calculating the shuffled vector sum 1000 times for the null distribution. We set a threshold $p$-value of 0.05 and we obtained in this manner 91 inverting cells out of 664 (14%). We obtained further confirmation of our cell typing above by corroborating that the detected cells with this method belonged to the direction selective groups reported obtained (groups 'DS', Baden et al.[2]).

## Ganglion cell modeling criteria

For the 4 mice retinas and 2 axolotl retinas where the set of unperturbed images was presented, we selected the cells to be modeled under the following criteria: a) they did have a classical STA, b) for mouse recordings, whether their STA was stable across the experiment (assessed by repeating the checkerboard stimuli at the end); c) the spiking responses across the unperturbed, repeated images were stable. To assess stability, we used the criterion of Cadena et al.[13] and discarded the neurons that showed a ratio of explainable-to-total variance smaller than 0.10. We were able to model 90 cells in mice and 50 cells in axolotl.

## Linear-nonlinear model (LN)

We implemented a regularized LN model. This model is fitted for each neuron separately and consists of: 1) a linear filter of weights $w_{ij}$ that compute a dot product with the input images (where $i$ and $j$ index space), 2) a pointwise nonlinear function $f_\theta$ that converts the filter output into a non-negative spike rate, and c) a Poisson noise model for

training. We chose $f_\theta$ to be a parametric softplus function such that $f_\theta(x) = \alpha ln(1 + \exp(\beta x + \gamma))$, where $\theta = \{\alpha, \beta, \gamma\}$. The spiking rate $r$ of a neuron given an input image $X$ follows as: $r(X) = f_\theta(\sum_{ij} w_{ij} x_{ij})$, where $X = x_{ij}$.

**Regularization.** two types of regularization were applied: a) $L_1$ regularization to induce sparsity: $L_{L_1} = \sum_{ij} W_{ij} V$; b) Laplacian regularization to induce smoothness: $L_\triangle = \sqrt{\sum_{ij}(W^* \triangle)^2_{ij}}$ where $W = w_{ij}$ and $\triangle = \frac{1}{4} \begin{pmatrix} 1 & 2 & 1 \\ 2 & -12 & 2 \\ 1 & 2 & 1 \end{pmatrix}$.

## Convolutional neural network model (CNN)

The first layer of the CNN model was a convolutional layer with four two-dimensional convolutional kernels, where the filters were learned from the data. For each experiment, the four kernels were common to all the modelled cells. The second layer was composed of one filter for each cell followed by a non-linearity. The weights of each of these filters were factorized in a two-dimensional spatial mask and a vector of feature weights (with one weight for each of the features extracted by the first layer) to decrease the number of model parameters, following previous work[12,13].

The model is composed of: a) convolutional kernel weights $k_{rsk}$ that compute convolutions with the input images (where $r$ and $s$ index the single kernel spatial dimensions and $k$ indexes kernels), b) pointwise nonlinear functions $f_{\theta_k^{[1]}}$ that convert the convolutional outputs into non-negative activation values. And in addition, for each neuron $n$: c) readout weights $w_{ijkn}$ which can be factorized as $w_{ijkn} = u_{ijn} v_{kn}$ where $i$ and $j$ index space, $u_{ijn}$ represent the spatial weights and $v_{kn}$ the feature weights, d) a pointwise nonlinear function $f_{\theta_n^{[2]}}$, and e) a Poisson noise model for training. We choose $f_{\theta_k^{[1]}}$ and $f_{\theta_n^{[2]}}$ to be softplus functions. The functions $f_{\theta_k^{[1]}}$ only had the parameter $\delta$ and $f_{\theta_n^{[2]}}$ had no parameters and were used like smoothed ReLus.

The outputs of the $k$th unit of the first layer were: $A_k = f_{\theta_k^{[1]}}(K_k * X)$. The spiking rate $r_n$ of the $n$th neuron given an input image $X$ was: $r_n(X) = f_{\theta_n^{[2]}}(\sum_k \sum_{ij} u_{ijn} v_{kn} A_{ijk})$. Additionally, batch normalization was applied to the outputs of the first layer.

**Regularization.** We used a Laplacian regularization on the convolutional kernels of the first layer, and we used $L_1$ regularization on the spatial weights and on the feature weights of the second layer such that: $L_\triangle = \frac{\sum_k \sum_{rs}(K_{rsk} * \triangle)_{rs}}{\varepsilon + \sum_{rsk}(K_{rsk})^2}$ and $L_{L_1} = \lambda_{sp} \sum_{ijn} |u_{ijn}| + \lambda_f \sum_{kn} |v_{kn}|$ with $\varepsilon = 10^{-8}$.

## Model training and cross-validation

**Model fitting.** Considering the $n$ recorded image-response pairs $(X_1, y_1), \ldots, (X_n, y_n)$, the resulting loss function is given by: $L = (\frac{1}{n}\sum_{k=1}^n r(X_k) - y_k log r(X_k)) + L_{L_1}\lambda_{L_1} + \lambda_\triangle L_\triangle$ where the first term corresponds to the negative log-likelihood of the Poisson loss and where $\lambda_{L_1}$ and $\lambda_\triangle$ are the hyperparameters which control the importance of the Laplacian and $L_1$ regularization terms.

We fitted the model by minimizing this loss using the Adam optimizer on the training set. The training set was composed of 2910 natural images and the associated responses (i.e., number of spikes elicited between the 30 ms and 350 ms after the onset of the flash). The batch size was fixed to 64. We used a learning rate initially equal to 0.001 and added a decay. In addition, we used early stopping. These two mechanisms prevented overfitting.

We cross-validated the hyperparameters $\lambda_{L_1}$ and $\lambda_\triangle$ for each neuron independently by performing a random search. The optimal hyperparameter values where the ones whose model produced the lowest loss value (without regularization terms) on the validation

dataset. When fitting models with different hyperparameters, we used the same split of data for training, validation, and testing across models.

**Model evaluation.** To evaluate the performance of the models, we used a testing set of 30 different stimuli where each stimulus has been repeated 30 times for mice or 20 times for axolotl. These repetitions allowed to separate prediction error in two parts: the error due to the limitations of the model and the error due to the intrinsic noise in the response.

Given $n$ responses to the same stimulus, $y_1, \dots, y_n$, we averaged them over odd and even numbered trials to get two estimates of the actual mean response, $\bar{y}_0$ and $\bar{y}_e$. We defined the reliability as the correlation between these estimates, $r_{\bar{y}_0, \bar{y}_e}$. Then, given the prediction of one model, $\hat{y}$, we estimated the noise-corrected correlation: $r_{nc} = \frac{\frac{1}{2}\left(r_{\hat{y}, \bar{y}_0} + r_{\hat{y}, \bar{y}_e}\right)}{\sqrt{r_{\bar{y}_0, \bar{y}_e}}}$ as introduced by Keshishian et al.[27]. We reported as model performance the corresponding noise-corrected R-squared, $R_{nc}^2 = r_{nc}^2$.

**LSTA prediction.** Given a model which predicts the firing rate $r(X)$ of a neuron in response to image $X$, we predicted the LSTA with: $LSTA(X) = \frac{\partial r(X)}{\partial X} = \left(\frac{\partial r(X)}{\partial x_{ij}}\right)$, the gradient of the model output with respect to the input image. This quantity is also referred to as the data Jacobian matrix[32]. Since all our models were implemented with TensorFlow[33], we took advantage of automatic differentiation to calculate the LSTAs.

**Vector field analysis**
To understand the relationship that a ganglion cell presents between the reference images and their corresponding LSTAs, we used the CNN model to predict for each modelled cell the LSTAs for 2910 images in our dataset. A principal component analysis was made on these predicted LSTAs on a cell by cell basis to obtain a two dimensional space composed by the two first principal components (PCA1 and PCA2). We projected each image as a point in the space defined by these two components, by calculating the dot product between the image and the two components. For each image we also made the projection of their corresponding LSTA. This produced a vector field for each cell. To ease the visualization of the vector field, the projection space was binned and the images and LSTAs falling inside each bin were averaged to be represented by a single point (the projected image) and arrow (the corresponding LSTA).

**Simple contrast model**
A simple contrast model was produced by assuming that the firing rate of the cell was given by: $r(I) \propto \sum_{ij} a_{ij}(l_{ij} - l_0)^2$, where $r(I)$ is the firing rate of a cell responding to the image $\mathbf{I}$, $l_{ij}$ are pixel values of the image, $l_0$ is a reference value of these pixels against which the contrast is calculated, and $\alpha_{ij}$ is a set of weighting parameters that should determine the contribution of each pixel. To reduce the number of parameters, instead of fitting one $\alpha_{ij}$ for each $l_{ij}$, we assumed the weights to be distributed as a 2D gaussian $\{\alpha_{ij}\} = A^* N((X - \mu) \bullet R(\theta), \sigma)$ with $\mathbf{X} = (x, y)$, $\mathbf{\mu} = (x_0, y_0)$, $\sigma = (\sigma_x, \sigma_y)$ and where R is the rotation matrix. The parameters $\gamma = \{x_0, y_0, \sigma_x, \sigma_y, \theta, A, l_0\}$ were fitted to the data. The fit was made through gradient descent so as to make the contrast model reproduce as well as possible the LSTAs predicted by the CNN (black arrows in Fig. 4e, f).

**Reporting summary**
Further information on research design is available in the Nature Research Reporting Summary linked to this article.

## Data availability
A minimum dataset (data from one example mouse experiment) generated in this study has been deposited in the Zenodo database at the link https://zenodo.org/record/6868362#.YtgeLoxBxH4. Due to the size of the full dataset, it has not been entirely uploaded in a public repository. Authors are happy to share it upon request. Also, Source data for the figs. panels 1e, f, g, 2b, e, 3e, 4g are provided with this paper. Source data are provided with this paper.

## Code availability
Modeling package is at https://gitlab.com/samuele_virgili/retina_modelling_2. The rest of the analysis code is available at https://github.com/samuelevirgili/Context-dependent-selectivity-to-natural-images-in-the-retina.

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

## Acknowledgements

We would like to thank Matthew Chalk for critical reading of the manuscript, Yannick Andéol for providing the axolotls and Dominic Gonschorek for his advice on the cell typing procedure. This work was supported by ANR grants (ANR-18-CE37-0011 – DECORE and ANR-20-CE37-0018-04- Shooting Star) and one from Retina France to O.M. and a grant from AVIESAN-UNADEV to O.M. and U.F.; U.F. was also supported by the Programme Investissements d'Avenir IHU FOReSIGHT 497 (ANR-18-IAHU-01), a grant from Sorbonne Université (CrInforNet) and a grant by the Agence Nationale de la Recherche (ANR-21-CE37-0024 NatNet-Noise). S.V. was supported by the European Union's Horizon 2020 research and innovation programme under the Marie Skłodowska-Curie grant agreement No 861423. M.G. was funded by a fellowship from Fondation de France. B.L. was funded by fellowships from PSL-ITI and Fondation pour la Recherche Médicale (FRM).

## Author contributions

M.G., B.L., S.V., T.M., U.F., and O.M. designed research. B.L. performed axolotl experiments. M.G. performed mice experiments. B.L. and S.V. did modeling.; M.P.V.C. did the temporal model; M.G., B.L., S.V., A.E., T.M., U.F., and O.M. analyzed the data and contributed to the writing of the manuscript. O.M. supervised research.

## Competing interests

The authors declare no competing interests.
