## [Peer Review File · Nature Communications]

Context-dependent selectivity to natural images in the retinaREVIEWER COMMENTS

Reviewer #1 (Remarks to the Author):

The manuscript by Goldin, Lefebvre, and Colleagues investigates the response properties of retinal ganglion cells (RGCs) to natural scene stimuli. They find a population of RGCs in mouse and salamander retinas that respond to local contrast. These RGCs do not have a feature selectivity that is fixed in polarity – responding to either increments (ON) or decrements (OFF) of light intensity. The authors show the responses of these RGCs are poorly predicted by a standard LN model for RGC responses. Instead, the responses of these RGCs are much better predicted by a convolutional neural network (CNN) that combines ON and OFF inputs from a hidden layer in the network. This hidden layer approximates that function of ON and OFF bipolar cells, interneurons between photoreceptors and RGCs. From a combination of CNN modeling, stimulus manipulations and measuring RGC responses on a multi-electrode array, the authors nicely demonstrate that these RGCs with a context-dependent spike-triggered receptive field, are reporting local contrast.

This is a very interesting study, well written, with a clear result. While a few groups have already been using CNNs in interesting ways to understand retinal responses, this manuscript does an excellent job of using the CNNs to learn and understand new features of RGC physiology. I particularly liked the L-AP4 experiment (Fig 3) and the image contrast reversal experiment (Fig 5).

I have a few relatively minor comments.

1. The authors emphasize the importance of natural scenes for the contrast signaling of the RGCs. This was not well demonstrated in the manuscript. Are natural scenes really required? Could the effect be reproduced just as easily with pink noise and a checkerboard pattern superimposed? It seems that all that is really required is an increase in local contrast and this could have been discovered with artificial stimuli given the right kind of experiment. Related, do the CNNs need to be trained on natural scenes to uncover this effect? It seems possible that the CNN could uncover the combination of ON and OFF inputs if trained on white noise or pink noise. Showing the CNN doesn't do this, would be a nice way to show there is something special about natural scenes per se – which is pretty heavily emphasized in the abstract and introduction.

2. Quantification of cells. The fraction of cells showing these effects was nicely reported early in the manuscript but fell off a bit toward the end. For example. How many cells were involved in the L-AP4 experiment and how many showed these effects? Did any fail to show these effects? Similarly, in figure 5, there are some example cells, but no quantification across the population – or perhaps I missed it? A bit more detail and completeness on reporting these effects across the population should be included.

3. Any information that could be added about cell types that did or did not show these effects would be useful. For example, I imagine the authors can identify putative alpha cells in their mouse MEA recordings. Do these cells demonstrate these effects? What about direction selective cells... these are relatively quick and easy to screen for, so if they authors have any information on them, that would be great. Even if the typology isn't fully worked out, I think future investigators would benefit from knowing cell types that do or don't perform these local-contrast computations.

Reviewer #2 (Remarks to the Author):

Goldin, Lefebvre and colleagues recorded from the retinas of mice and axolotl and found that the image-evoked responses approximated a computation of local contrast by each cell. Previous studies have found that CNN models outperform conventional LN models at predicting retinal responses, but

none have provided a simplified explanation of the nonlinearities produced by the CNN. This conclusion is supported by their recordings using a novel stimulus set of perturbations. This work, although impressive, could be aided by a bit more explanation for the reader.

* Can you comment more on the statistics of the ganglion cells you record? Is there any way to know (with static images) whether or not you are recording non-ON/OFF ganglion cells like direction-selective ganglion cells? These cells can also have non-stereotypic responses (e.g. see Ding, ... , Wei, eLife, 2021).

* How could your model account for the differences in latencies of responses that are observed? Is there a simple way to change it to make it perform this way?

* Figure 3 explains the requirement of both the ON and OFF maps. However the convolutional model has 4 kernels. What do the other 2 kernels look like? Can you show them perhaps in a supplementary figure to give the reader a better idea of what's being computed?

* Similarly the feature weights of the CNN are not explored beyond the first 2 in Figure 3b. Within this 4D space are there clusters of cells with similar weights, like Figure 4d vs 4e cells? Or is there more of a continuum? If there are clusters, do these clusters vary across retinotopic space?

* Commonly a scatter plot of R^2 is shown across models, where each cell is a point, rather than reporting the mean error alone. Could you scatter plot R^2 of LN vs CNN vs contrast model? This will give the reader the sense of the number of cells better explained vs if it's a few outlier cells (this seems unlikely but would be useful to show).

Reply to reviewers

We have put in **blue** our replies, and in **red** the modifications we made in the text and the figures that were added.

Response to Reviewer #1:

Reviewer: The manuscript by Goldin, Lefebvre, and Colleagues investigates the response properties of retinal ganglion cells (RGCs) to natural scene stimuli. They find a population of RGCs in mouse and salamander retinas that respond to local contrast. These RGCs do not have a feature selectivity that is fixed in polarity – responding to either increments (ON) or decrements (OFF) of light intensity. The authors show the responses of these RGCs are poorly predicted by a standard LN model for RGC responses. Instead, the responses of these RGCs are much better predicted by a convolutional neural network (CNN) that combines ON and OFF inputs from a hidden layer in the network. This hidden layer approximates that function of ON and OFF bipolar cells, interneurons between photoreceptors and RGCs. From a combination of CNN modeling, stimulus manipulations and measuring RGC responses on a multi-electrode array, the authors nicely demonstrate that these RGCs with a context-dependent spike-triggered receptive field, are reporting local contrast.

This is a very interesting study, well written, with a clear result. While a few groups have already been using CNNs in interesting ways to understand retinal responses, this manuscript does an excellent job of using the CNNs to learn and understand new features of RGC physiology. I particularly liked the L-AP4 experiment (Fig 3) and the image contrast reversal experiment (Fig 5).

Authors: We thank the reviewer for their appreciation of our work.

Reviewer: I have a few relatively minor comments.

Remark 1:

a) The authors emphasize the importance of natural scenes for the contrast signaling of the RGCs. This was not well demonstrated in the manuscript. Are natural scenes really required? Could the effect be reproduced just as easily with pink noise and a checkerboard pattern superimposed? It seems that all that is really required is an increase in local contrast and this could have been discovered with artificial stimuli given the right kind of experiment. Related, do the CNNs need to be trained on natural scenes to uncover this effect? It seems possible that the CNN could uncover the combination of ON and OFF inputs if trained on white noise or pink noise. Showing the CNN doesn't do this, would be a nice way to show there is something special about natural scenes per se – which is pretty heavily emphasized in the abstract and introduction.

Authors: We apologize for the confusion. We did not want to claim that natural scenes were necessary to see this effect, and we agree that a carefully chosen artificial stimulus could probably give the same result: in fact, in our figure 5, we showed LSTA polarity inversion for negative images, which are not natural.

Our intent was rather to emphasize the importance of probing the selectivity of ganglion cells during natural scene stimulation: if selectivity is tested with artificial stimuli like white noise, it is not clear that ganglion cells will be selective to the same feature when stimulated with natural scenes. This is shown in our data, since ganglion cells that appear OFF when probed with a checkerboard can become ON when probed with our method, depending on the natural image presented. Our method allows probing selectivity directly during natural stimuli, and thus avoids relying on extrapolating results from artificial stimuli to natural ones. We have clarified this in the text.

Related to that, we don't think there is any evidence to claim that we need to train the CNN model on natural images. It could be that training a model on either checkerboard or pink noise data would allow predicting our results. However, in practice, this is a very hard challenge since we would need to have a model that can generalize from checkerboard to natural image statistics, to be able to predict responses to natural images and the corresponding LSTAs. Previous works (e.g. Heitman et al, 2016) have shown that achieving this generalization is an open challenge even in the retina. This is also why we present here a strategy that avoids to solve this problem by directly probing selectivity during natural scene stimulation. We thus think that finding a strategy that achieves this generalization goes beyond the scope of this study. We have changed the abstract and introduction to answer the concerns of the reviewer and clarify our purposes.

Changes:

i) We have modified slightly the abstract to remove any misunderstanding about this point. It now reads as follows:

“Retina ganglion cells extract specific features from natural scenes and send this information to the brain. In particular, they respond to local light increase (ON responses), and/or decrease (OFF). However, it is unclear if this ON-OFF selectivity, characterized with synthetic stimuli, is maintained under natural scene stimulation. Here we recorded ganglion cell responses to natural images slightly perturbed by random noise patterns to determine their selectivity during natural stimulation. The ON-OFF selectivity strongly depended on the specific image. A single ganglion cell can signal luminance increase for one image, and luminance decrease for another. Modeling and experiments showed that this resulted from the non-linear combination of different retinal pathways. Despite the versatility of the ON-OFF selectivity, a systematic analysis demonstrated that contrast was reliably encoded in these responses. Our perturbative approach uncovered the selectivity of retinal ganglion cells to more complex features than initially thought.”

ii) We modified also some sentences in the Introduction, which now reads as follows:

“To carry out complex visual tasks, it has been proposed that each area of the visual system extracts different features from the visual scene. The complexity of these extracted features increases along the hierarchy of visual areas (Yamins & Dicarlo, 2016). This feature extraction process starts in the retina, whose output is composed of retinal ganglion cells. These cells can be classified in different cell types (Baden et al., 2016) and each of them is supposed to extract a low-level visual feature from the visual scene (Azeredo da Silveira & Roska, 2011). This feature selectivity is often inferred by reverse correlation using a white noise stimulus (Chichilnisky, 2001). This type of linear response modeling using stimuli with simple statistics, although effective in determining an approximate receptive field of the cell, falls short of modeling the full nonlinear response profile of the retina to complex stimuli such as the ones it is usually exposed in nature.

A basic example of feature extraction is the selectivity of ganglion cells to luminance increase or decrease: the so called ON cells are those sensitive to light increments, whereas OFF cells are those sensitive to light decrements, and ON-OFF cells respond to both. Recent works have shown that, surprisingly, this ON-OFF selectivity depends on the visual context. For example, it changes with the background luminance (Pearson & Kerschensteiner, 2015; Tikidji-Hamburyan et al., 2015). Transient variations of the preferred polarity have also been reported in response to large motion in the periphery (Geffen, De Vries, & Meister, 2007). In these studies, retinal processing was probed with simple artificial stimuli. It is not clear how ON-OFF selectivity would generalize to more complex stimuli, even in a regime of constant average global contrast and luminance (Heitman et al, 2016).

Here we use a novel perturbative approach for probing context-dependent selectivity with perturbations added on top of natural scenes. We stimulated ganglion cells of mouse and axolotl retinas with natural images, and then added small checkerboard-like perturbations on top of them. These perturbations evoked small changes on the responses of retinal ganglion cells. When analyzing systematically the responses to these perturbations, we found that the same ganglion cell can be selective to light increments when the perturbations are added on top of one natural image, and to light decrements when they are added on top of another. Ganglion cells can thus switch their selectivity from ON to OFF depending on the context, and do so during natural scene stimulation. We designed a non-linear model to explain and predict these changes, and mapped this model to specific circuits in the retinal network. Finally, we demonstrated that this strong context dependence is compatible with a robust computation of a more abstract feature: contrast.”

iii) We added this sentence in the end of the second paragraph of the Discussion:

“Note that it is also possible that this polarity inversion could also be observed for stimuli other than natural images. For example, we observed it for negative images (Fig. 5).”

Remark 2:

Quantification of cells. The fraction of cells showing these effects was nicely reported early in the manuscript but fell off a bit toward the end. For example,

a) How many cells were involved in the L-AP4 experiment and how many showed these effects? Did any fail to show these effects?

Authors: We agree that the analysis was incomplete and we have added a full population analysis of these effects.

Changes:

i) We added these sentences to the second paragraph of the “Polarity changes...” subsection of the Results section (page 13, line 10) as follows:

“We detected across 3 mice retinas a total of 209 cells showing LSTAs before L-AP4 application. Of these, 26 were detected to invert polarity.”

“In only in 4% of the cases, LSTAs with ON polarity remained after L-AP4 application, while LSTAs with OFF polarity did stay in 26% of the cases (Fig. 3E). The decrease of LSTAs with OFF polarity is due to the experiment duration when adding L-AP4 and measuring again LSTAs: ganglion cells tend to lose sensitivity after many hours of recording, and this results in some LSTAs disappearing over time. Nevertheless, the difference in decrease between the LSTAs showing ON and OFF polarity is highly significant ($p=2 \times 10^{-3}$, Fisher exact test).”

ii) We added to Fig. 3E caption:

“The difference is highly significant ($p=2 \times 10^{-3}$, Fisher exact test).”

b) Similarly, in figure 5, there are some example cells, but no quantification across the population – or perhaps I missed it? A bit more detail and completeness on reporting these effects across the population should be included.

Authors: We agree that this was incomplete. We performed more experiments containing the negative images stimulus.

In these experiments we recorded a total of 527 cells that showed LSTAs, and of these, 39 cells (8%) showed LSTA polarity inversion. Among the 39 inverting cells, 23 (59%) were observed to reliably change their LSTA polarity when presented with negative natural images.

Changes:

i) We modified the last paragraph of the subsection “Changes in polarity...” in the Results section as follows (page 20, line 14):

“Among the cells detected to invert (39 out of 527 cells recorded in 4 mouse retinas), 57±9% were observed to reliably change their LSTA polarity when presented with negative natural images (Fig. 5B).”

Remark 3:

Any information that could be added about cell types that did or did not show these effects would be useful. For example, I imagine the authors can identify putative alpha cells in their mouse MEA recordings. Do these cells demonstrate these effects? What about direction selective cells... these are relatively quick and easy to screen for, so if they authors have any information on them, that would be great. Even if the typology isn't fully worked out, I think future investigators would benefit from knowing cell types that do or don't perform these local-contrast computations.

Authors: We agree with the reviewer that investigating and reporting the cell types that present the polarity inverting behavior can be of great utility for the community. For this reason, we made three further experiments in mice retinas and applied the same perturbation protocol to reveal polarity inversions, together with two additional typing protocols:

The typing protocols are inspired by the work of Baden et al. 2016. In the first one, we applied a “chirp” stimulus (full field luminance ON and OFF steps, plus varying full field frequencies and amplitudes) which was used to cluster the neuronal responses according to the 32 cell types provided in their work. The second typing protocol consists of applying drifting gratings in 8 angular directions and calculating significantly tuned neurons to any specific direction.

We found that the inverting cells in these experiments do not belong to any specific type, according to the work of Baden et al. 2016, but can appear in several groups; and only one among them presented a direction selectivity to the drifting grating. To answer the particular inquiry of the reviewer, there were no inverting cells belonging to the ON alpha type, nor to the OFF alpha sustained type. This is because these cells are pure ON or OFF. However, we found OFF alpha transient, OFF mini alpha transient and ON mini alpha inverting cells. These later types, on the contrary, may contain ON and OFF responses (see **Supplementary Fig. 3**).

Changes:

i) We include a New Methods section:

Cell typing

We performed three further experiments in mice retinas of C57BL6J mice of 17 weeks to find out if our polarity inverting cells do belong to any specific cell type.

Stimulus: In addition to our perturbation protocol to detect polarity inverting cells, we applied two additional ones. 1) A full field 'chirp' stimulus composed of ON and OFF steps, plus varying full field frequencies and amplitudes, with luminance values ranging from 0 to 1. The stimulus is the exact same that Baden et al. 2016 used to find and classify 32 different types of ganglion cells. It was played at 50Hz, containing 20 repetitions of 32 s length, (See Supplementary Fig. 3). 2) Drifting gratings (DG) moving in 8 different directions with a speed of 479.5 $\mu\text{m/s}$, at a spatial period of 959 μm (274 pixels at 3.5 $\mu\text{m/pix}$) and at 50% Michelson contrast (0.75-0.25 luminance). Each DG lasts 10 s, preceded by 2 s of gray (0.5 luminance), the temporal period being 2 s. Therefore, each grating edge goes through a ganglion cell's receptive fields 5 times per DG. The 8 directions were repeated 4 times in a pseudo-random manner. The stimulus profile and dynamics is identical to the one described in Yao et al. 2018 to retrieve direction selective cells. In our case we used a unique luminance value, as described in the Visual stimulation section above.

Typing: To cluster cells in different types, we based our analysis on the chirp and checkerboard stimulus responses, and representing each ganglion cell with a reduced representative vector. To obtain these vectors, first we constructed peri-stimulus histograms (PSTH) from the spikes evoked from the chirp stimulus, using a binning of 100 ms. Then, for each experiment, we z-scored all PSTHs and performed a PCA on them. We kept the number of components that were needed to explain 80% variance of the data (around 12 components). Second, we used the temporal profile (21 samples at 30 Hz) of each cell's STA obtained using the checkerboard stimulus. We z-scored it and performed a PCA, keeping the first component, which explains around 60% of the variance. This adds information about the classical STA polarity of the ganglion cells. Third, we used the area of the ellipse fitted to the classical STA, as the product value of their major and minor axis σ values. These areas were normalized from 0 to 1. In this way, we obtain a data vector of around 14 values, depending on the experiment, that describes each ganglion cell according to their response to a chirp and a checkerboard. Then, we performed an agglomerative clustering, setting the threshold value in a way that all clusters look homogeneous across PSTHs and STAs. This resulted in overclustering that produced around 50 ganglion cell groups (from around 200 cells in each experiment). In the last step, we assigned each cluster group to one of the 32 types described in Baden et al. 2016. To do this, we used the Calcium imaging data provided by the authors to match it with our data. We based ourselves in their Extended data Figure 1, where the authors link electrophysiology and calcium imaging by means of a convolution between a Ca^{2+} event triggered by a single spike. We transformed our PSTHs by convolving them with a decaying exponential, in which we adjusted the temporal decay constant to maximize correlation of our cluster groups and theirs (median maximum correlation of 0.76). Cell types that present strong responses to the modulating frequencies and amplitude were assigned correctly, while other types which mostly respond to ON/OFF steps, were assigned in a second round of correlation match after excluding the former groups. Besides the correlation of the chirp traces, we confirmed the correct assigning of groups by checking that the ellipses of each type form a proper mosaic, that the spatial STAs look uniform, the similarity of their direction selectivity PSTHs (see below) and that of the spike waveforms. Finally, we computed a correlation matrix between the average chirp response of each type to show that the groups are homogeneous (Supplementary Fig. 3A).

Direction selectivity: We constructed PSTHs from the spikes evoked from the chirp stimulus, and calculated the mean firing rate evoked by each DG direction, and normalized it to the maximum direction for each cell (values 0 to 1). To assess selectivity, we calculated the vector sum of these normalized response vectors, which spanned values from 0 to 2, as it is usually done (Baden et al. 2016, Dao et al. 2018). To test if a neuron was direction selective, we performed a shuffle test on the cells whose resulting vector sum was bigger than 0.5, randomly permuting the direction labeling of every DG trial, and calculating the shuffled vector sum 1000 times for the null distribution. We set a threshold p-value

of 0.05 and we obtained in this manner 91 inverting cells out of 664 (14%). We obtained further confirmation of our cell typing above by corroborating that the detected cells with this method belonged to the direction selective groups reported obtained (groups 'DS', Baden et al. 2016).

ii) In the "Ganglion cells can change..." subsection of the Results section we add in page 7 line 11:

"We looked further into the cell classification using two stimuli, a chirp (Baden et al, 2016) and drifting gratings (Yao et al. 2018) (see Methods and Supplementary Fig. 3). We found that the inverting cells in these experiments do not belong to a specific type, but can appear in several groups. We found that all cells showing polarity inversion had at least a detectable response to both ON and OFF flashes. Cell types with a "pure" ON or OFF selectivity did not show polarity inversion. However, the ratio between the ON and the OFF responses could vary very broadly: some cells (e.g. the ON transient type) had a strong ON response and a barely detectable OFF response to flashes, but showed polarity inversion for some cells when probed with natural images (see Supplementary Fig. 3, and Supplementary Table 1). Only one inverting cell was direction selective."

iii) We include new Supplementary Fig. 3:

Supplementary Figure 3. Ganglion cell types

(A) Correlation matrix showing homogeneity of the clustered cells. Each cell within a group was correlated with all other 29 cell type groups average PSTH we found (see Supplementary Table 1). The matrix shows the average of this values.

(B) Example cells that showed polarity inversion. All cells contain a detectable ON and OFF component in their responses, which is seen at the onset of the ON light step and OFF light step. Cells with pure ON or OFF responses do not show polarity inversion. Grayscale for spatial STAs denote polarity, dark are OFF and bright are ON response. Scale bar: 200 μ m.

(C) Example data set from a cell type group that was used to assess the homogeneity of the clustered cell types. This group, the ON mini alpha transient, shows a clear mosaic, overlapping Temporal STAs (gray curves), and absence of directional tuning (see Methods). On the contrary groups with undetectable OFF responses, (e.g., the ON alpha type, with pure ON responses) did not have any polarity inverting cell. The same applied to groups with undetectable ON responses (pure OFF). Scale bar: 200 μ m.

(D) The polar plots show with blue lines the vector sum of the normalized firing rate responses of cells to drifting gratings in 8 directions (see Methods). Left: All ganglion cells where we found polarity inversion. Right: All ganglion cells that were detected to be directionally selective (see Methods).

ii) We include new Supplementary Table 1:

Group Type	# Cells detected	# Cells inverting	% Cells inverting
OFF local OS	21	6	29%
OFF DS	0	-	-
OFF step	15	0	0%
OFF slow	33	1	3%
OFF alpha sustained	19	0	0%
ON-OFF JAM-B	21	0	0%
OFF sustained	9	0	0%
OFF alpha transient	13	3	23%
OFF mini alpha transient	31	6	19%
ON-OFF local-edge W3	21	2	10%
ON-OFF local	18	1	6%
ON-OFF DS 1	14	1	7%
ON-OFF DS 2	21	0	0%
ON-OFF local OS	33	4	12%
ON step	28	3	11%
ON DS transient	7	0	0%
ON local transient OS	34	0	0%
ON transient	19	3	16%
ON transient large	0	-	-
ON high frequency	13	1	8%
ON low frequency	26	1	4%
ON sustained	27	2	7%
ON mini alpha	27	2	7%
ON alpha	21	0	0%
ON DS sustained 1	2	0	0%
ON DS sustained 2	0	-	-
ON slow	37	0	0%
ON contrast suppression	8	0	0%
ON DS sustained 3	3	0	0%
ON local sustained OS	16	0	0%
OFF suppression 1	13	0	0%
OFF suppression 2	6	0	0%
Total number of cells	556	36	6%

Supplementary Table 1. Polarity inverting can belong to many different cell types.

Cell type classification from the chirp stimulus and the checkerboard stimulus (See Methods). The polarity inverting cells do not belong to a particular cell type, but may belong to many. We found polarity inverting cells in 14 out of 29 types of cells detected. The lower percentage of total inverting cells reported here is because we take into account all the cells that respond to the typing stimulus, including the ones that do not present LSTAs to our perturbative approach.

Response to Reviewer #2:

Reviewer: Goldin, Lefebvre and colleagues recorded from the retinas of mice and axolotl and found that the image-evoked responses approximated a computation of local contrast by each cell. Previous studies have found that CNN models outperform conventional LN models at predicting retinal responses, but none have provided a simplified explanation of the nonlinearities produced by the CNN. This conclusion is supported by their recordings using a novel stimulus set of perturbations.

Authors: We thank the reviewer for their appreciation of our work.

We provide below the answer to the 5 remarks of Reviewer #2:

Reviewer: This work, although impressive, could be aided by a bit more explanation for the reader.

Remark 1:

*** Can you comment more on the statistics of the ganglion cells you record? Is there any way to know (with static images) whether or not you are recording non-ON/OFF ganglion cells like direction-selective ganglion cells? These cells can also have non-stereotypic responses (e.g see Ding, ... , Wei, eLife, 2021).**

Authors: Since this remark is similar to the Remark #3 of Reviewer #1 that we answered above, we repeat our response here. Regarding the comment on the use of static images for this purpose, we could not do a proper typing relying on them, so we performed new experiments for that.

Copy of Response to Reviewer 1, Remark 3: We agree with the reviewer that investigating and reporting the cell types that present the polarity inverting behavior can be of great utility for the community. For this reason, we made three further experiments in mice retinas and applied the same perturbation protocol to reveal polarity inversions, together with two additional typing protocols:

The typing protocols are inspired by the work of Baden et al. 2016. In the first one, we applied a “chirp” stimulus (full field luminance ON and OFF steps, plus varying full field frequencies and amplitudes) which was used to cluster the neuronal responses according to the 32 cell types provided in their work. The second typing protocol consists of applying drifting gratings in 8 angular directions and calculating significantly tuned neurons to any specific direction.

We found that the inverting cells in these experiments do not belong to any specific type, according to the work of Baden et al. 2016, but can appear in several groups; and only one among them presented a direction selectivity to the drifting grating. To answer the particular inquiry of the reviewer, there were no inverting cells belonging to the ON alpha type, nor to the OFF alpha sustained type. This is because these cells are pure ON or OFF. However, we found OFF alpha transient, OFF mini alpha transient and ON mini alpha inverting cells. These later types, on the contrary, may contain ON and OFF responses (see Supplementary Fig. 3).

Changes:

i) We include a New Methods section:

Cell typing

We performed three further experiments in mice retinas of C57BL6J mice of 17 weeks to find out if our polarity inverting cells do belong to any specific cell type.

Stimulus: In addition to our perturbation protocol to detect polarity inverting cells, we applied two additional ones. 1) A full field ‘chirp’ stimulus composed of ON and OFF steps, plus varying full field frequencies and amplitudes, with luminance values ranging from 0 to 1. The stimulus is the exact same that Baden et al. 2016 used to find and classify 32 different types of ganglion cells. It was played at 50Hz, containing 20 repetitions of 32 s length, (See Supplementary Fig. 3). 2) Drifting gratings (DG) moving in 8 different directions with a speed of 479.5 $\mu\text{m/s}$, at a spatial period of 959 μm (274 pixels at 3.5 $\mu\text{m/pix}$) and at 50% Michelson contrast (0.75-0.25 luminance). Each DG lasts 10 s, preceded by 2 s of gray (0.5 luminance), the temporal period being 2 s. Therefore, each grating edge goes through a ganglion cell’s receptive fields 5 times per DG. The 8 directions were repeated 4 times in a pseudo-random manner. The stimulus profile and dynamics is identical to the one described in Yao et al. 2018 to retrieve direction selective cells. In our case we used a unique luminance value, as described in the Visual stimulation section above.

Typing: To cluster cells in different types, we based our analysis on the chirp and checkerboard stimulus responses, and representing each ganglion cell with a reduced representative vector. To obtain these vectors, first we constructed peri-stimulus histograms (PSTH) from the spikes evoked from the chirp stimulus, using a binning of 100 ms. Then, for each experiment, we z-scored all PSTHs and performed a PCA on them. We kept the number of components that were needed to explain 80% variance of the data (around 12 components). Second, we used the temporal profile (21 samples at 30 Hz) of each cell’s STA obtained using the checkerboard stimulus. We z-scored it and performed a PCA, keeping the first component, which explains around 60% of the variance. This adds information about the classical STA polarity of the ganglion cells. Third, we used the area of the ellipse fitted to the classical STA, as the product value of their major and minor axis σ values. These areas were normalized from 0 to 1. In this way, we obtain a data vector of around 14 values, depending on the experiment, that describes each ganglion cell according to their response to a chirp and a checkerboard. Then, we performed an agglomerative clustering, setting the threshold value in a way that all clusters look homogeneous across PSTHs and STAs. This resulted in overclustering that produced around 50 ganglion cell groups (from around 200 cells in each experiment). In the last step, we assigned each cluster group to one of the 32 types described in Baden et al. 2016. To do this, we used the Calcium imaging data provided by the authors to match it with our data. We based ourselves in their Extended data Figure 1, where the authors link electrophysiology and calcium imaging by means of a convolution between a Ca^{2+} event triggered by a single spike. We transformed our PSTHs by convolving them with a decaying exponential, in which we adjusted the temporal decay constant to maximize correlation of our cluster groups and theirs (median maximum correlation of 0.76). Cell types that present strong responses to the modulating frequencies and amplitude were assigned correctly, while other types which mostly respond to ON/OFF steps, were assigned in a second round of correlation match after excluding the former groups. Besides the correlation of the chirp traces, we confirmed the correct assigning of groups by checking that the ellipses of each type form a proper mosaic, that the spatial STAs look uniform, the similarity of their direction selectivity PSTHs (see below) and that of the spike waveforms. Finally, we computed a correlation matrix between the average chirp response of each type to show that the groups are homogeneous (Supplementary Fig. 3A).

Direction selectivity: We constructed PSTHs from the spikes evoked from the chirp stimulus, and calculated the mean firing rate evoked by each DG direction, and normalized it to the maximum direction for each cell (values 0 to 1). To assess selectivity, we calculated the vector sum of these normalized response vectors, which spanned values from 0 to 2, as it is usually done (Baden et al. 2016, Dao et al. 2018). To test if a neuron was direction selective, we performed a shuffle test on the cells whose resulting vector sum was bigger than 0.5, randomly permuting the direction labeling of every DG trial, and calculating the shuffled vector sum 1000 times for the null distribution. We set a threshold p-value of 0.05 and we obtained in this manner 91 inverting cells out of 664 (14%). We obtained further

confirmation of our cell typing above by corroborating that the detected cells with this method belonged to the direction selective groups reported obtained (groups 'DS', Baden et al. 2016).

ii) In the "Ganglion cells can change..." subsection of the Results section we add in page 7 line 11:

"We looked further into the cell classification using two stimuli, a chirp (Baden et al, 2016) and drifting gratings (Yao et al. 2018) (**see Methods and Supplementary Fig. 3**). We found that the inverting cells in these experiments do not belong to a specific type, but can appear in several groups. We found that all cells showing polarity inversion had at least a detectable response to both ON and OFF flashes. Cell types with a "pure" ON or OFF selectivity did not show polarity inversion. However, the ratio between the ON and the OFF responses could vary very broadly: some cells (e.g. the ON transient type) had a strong ON response and a barely detectable OFF response to flashes, but showed polarity inversion for some cells when probed with natural images (see Supplementary Fig. 3, and Supplementary Table 1). Only one inverting cell was direction selective."

iii) We include new Supplementary Fig. 3:

Supplementary Figure 3. Ganglion cell types

(A) Correlation matrix showing homogeneity of the clustered cells. Each cell within a group was correlated with all other 29 cell type groups average PSTH we found (see Supplementary Table 1). The matrix shows the average of this values.

(B) Example cells that showed polarity inversion. All cells contain a detectable ON and OFF component in their responses, which is seen at the onset of the ON light step and OFF light step. Cells with pure ON or OFF responses do not show polarity inversion. Grayscale for spatial STAs denote polarity, dark are OFF and bright are ON response. Scale bar: 200 μ m.

(C) Example data set from a cell type group that was used to assess the homogeneity of the clustered cell types. This group, the ON mini alpha transient, shows a clear mosaic, overlapping Temporal STAs (gray curves), and absence of directional tuning (see Methods). On the contrary groups with undetectable OFF responses, (e.g., the ON alpha type, with pure ON responses) did not have any polarity inverting cell. The same applied to groups with undetectable ON responses (pure OFF). Scale bar: 200 μ m.

(D) The polar plots show with blue lines the vector sum of the normalized firing rate responses of cells to drifting gratings in 8 directions (see Methods). Left: All ganglion cells where we found polarity inversion. Right: All ganglion cells that were detected to be directionally selective (see Methods).

iii) We include new Supplementary Table 1:

Group Type	# Cells detected	# Cells inverting	% Cells inverting
OFF local OS	21	6	29%
OFF DS	0	-	-
OFF step	15	0	0%
OFF slow	33	1	3%
OFF alpha sustained	19	0	0%
ON-OFF JAM-B	21	0	0%
OFF sustained	9	0	0%
OFF alpha transient	13	3	23%
OFF mini alpha transient	31	6	19%
ON-OFF local-edge W3	21	2	10%
ON-OFF local	18	1	6%
ON-OFF DS 1	14	1	7%
ON-OFF DS 2	21	0	0%
ON-OFF local OS	33	4	12%
ON step	28	3	11%
ON DS transient	7	0	0%
ON local transient OS	34	0	0%
ON transient	19	3	16%
ON transient large	0	-	-
ON high frequency	13	1	8%
ON low frequency	26	1	4%
ON sustained	27	2	7%
ON mini alpha	27	2	7%
ON alpha	21	0	0%
ON DS sustained 1	2	0	0%
ON DS sustained 2	0	-	-
ON slow	37	0	0%
ON contrast suppression	8	0	0%
ON DS sustained 3	3	0	0%
ON local sustained OS	16	0	0%
OFF suppression 1	13	0	0%
OFF suppression 2	6	0	0%
Total number of cells	556	36	6%

Supplementary Table 1. Polarity inverting can belong to many different cell types.

Cell type classification from the chirp stimulus and the checkerboard stimulus (See Methods). The polarity inverting cells do not belong to a particular cell type, but may belong to many. We found polarity inverting cells in 14 out of 29 types of cells detected. The lower percentage of total inverting cells reported here is because we take into account all the cells that respond to the typing stimulus, including the ones that do not present LSTAs to our perturbative approach.

Remark 2:

* How could your model account for the differences in latencies of responses that are observed? Is there a simple way to change it to make it perform this way?

Authors: The CNN model we used in our paper only predicts firing rate and does not have a temporal dimension, so it cannot predict latencies: it takes each image as an input and predicts the firing rate as an output. To answer the reviewer, we have expanded our CNN model so that it can also take time into account, in the following way:

We kept the same structure for the first layer, i.e. several kernels convolved with the input, but we switched from 2D to 3D kernels (two spatial and one time dimension). For the second layer, we added a temporal mask to the spatial mask in order to integrate inputs over time and space (see **Supplementary Fig. 8** below).

Supplementary Figure 8. Extended CNN model to account for time.

(A) The modified CNN model: For the first layer it has several kernels as our original CNN, but we switch from 2D to 3D kernels (two spatial and one time dimension). For the second layer, we add a temporal mask to the spatial mask in order to integrate inputs over time besides space. We transform each input natural image into a movie composed of 10 successive frames, all identical to the starting natural image. The output for each cell is the image response spike count, divided in 30 ms time bins. Therefore, a single spike count number from our original CNN, which integrated 300 ms, becomes a vector with 10 entries in the new framework. We trained the model to predict spike counts at a given time bin using the input at previous time bins. The response latency was detected as the bin presenting the response onset, defined as the bin in which the first PSTH maximum was observed, after smoothing.

To learn this model we needed to restructure the input and output data. We transformed each input natural image into a movie composed of 10 successive frames, all identical to the starting natural image. The output for each cell is the image response spike count, divided in 30 ms time bins. Therefore, a single spike count number from our original CNN, which integrated 300 ms, becomes a vector with 10 entries in the new framework. We trained the model to predict spike counts at a given time bin using the input at previous time bins.

In the analysis, the response latency was detected as the bin presenting the response onset, defined as the bin in which the first PSTH maximum was observed, after smoothing. When tested on the repeated images test set, this model gave heterogeneous prediction performances. In 60% of the cases, the latency difference was predicted with an error below 60 ms.

Supplementary Figure 8. Extended CNN model to account for time.

(B) When tested on the repeated images test set, this model gave heterogeneous prediction performances. In 40% of the cases, the latency difference with the real data was predicted with an error above 60 ms. The performance of this model strongly depended on the cell, and for some cells, it made important errors.

However, the performance of this model strongly depended on the cell, and for some cells, it still made important errors. Improving the performance further would require to try very different model architectures (e.g. Vierock et al, 2021 ; McIntosh et al, 2016) and to learn the model on responses to natural movies, rather than flashed natural images. We think this is beyond the scope of this paper but we now comment about this in the discussion.

Changes:

i) We add Supplementary Figure 8 with this new model results on latency prediction.

ii) We add these sentences in the end of the third paragraph of the first Discussion subsection (page 22, line 13):

“In order to account for the different response latencies, we implemented a simple extended version of our CNN model to make it time dependent (see Supplementary Fig. 8). Although this simple modification predicted the response latency well for some cells, it made important mistakes for others. Improving the performance further would require to try very different model architectures (e.g. Vierock et al, 2021 ; McIntosh et al) and to learn the model on responses to natural movies, rather than flashed natural images.”

Remark 3:

* Figure 3 explains the requirement of both the ON and OFF maps. However the convolutional model has 4 kernels. What do the other 2 kernels look like? Can you show them perhaps in a supplementary figure to give the reader a better idea of what’s being computed?

Authors: We now show the four kernels found for our experiments in **Supplementary Fig. 5** We also provide an angular average of their sections, calculated from the kernel center, to provide to the reader detailed information about their surround. The convolutional kernels show either a clear ON or OFF polarity.

Supplementary Figure 5. Convolutional kernels and scatter plots of model features, in reference to Fig. 3.

(A) The four kernels of the model displayed in Fig. 3A

(B) Average profile calculated from the center of each kernel across every angle. The kernels display a clear opposing polarity surround.

Changes:

i) We added Supplementary Figure 5 with the full kernels information

ii) We added to the caption of panel 3A: (see Supplementary Figure 5A,B for more details).

Remark 4:

* Similarly the feature weights of the CNN are not explored beyond the first 2 in Figure 3b. Within this 4D space are there clusters of cells with similar weights, like Figure 4d vs 4e cells? Or is there more of a continuum? If there are clusters, do these clusters vary across retinotopic space?

Authors: We show here the clustering of one mouse experiment with 41 modelled cells in 4D. We could not obtain any clear cluster separation.

C

Supplementary Figure 5. Convolutional kernels and scatter plots of model features, in reference to Fig. 3.

(C) Scatter plots of model features 41 modeled cells for one example retina. The feature weights here correspond the kernels in A. Cells labeled with crosses are polarity inverting. For visualization purposes, the 4 features were scatter-plotted in pairs. The different colors represent different putative clusters that were obtained with standard clustering techniques. We could not obtain any clear cluster separation

In this experiment and in all the others cells do not seem to cluster neatly in the feature weight space and there does not seem to be a separation between the inverting and non-inverting ones. Therefore, it was not possible to find a relation between the position of a cell in the feature weight space and the shape of its arrow field. For the attempted clustering we used the algorithm k-means and the optimal number of clusters was assessed with both the silhouette and the knee method. We also tried to repeat the same clustering in 2D space by averaging the weights of the kernels with the same polarity but the result was equally inconclusive. The example reported above was nevertheless added to supplementary Figure 5 as we agree it might be helpful for the reader. It remains possible that clusters would appear if we had a very large number of cells (note that for example, Baden et al. could cluster ganglion cells in different types using a database of 10000 cells), but we think collecting such a large dataset for this purpose is beyond the scope of this study.

Changes:

i) We added a panel in Supplementary Figure 5

ii) We added a sentence at the end of the first paragraph of the “Polarity changes are due...” subsection of the Results section (page 12, line 26):

“We tried clustering the modeled cells in the 4D feature weight space, to see whether the inverting cells would cluster together. However, we found no clear patterns in the feature space (Supplementary Fig. 5C). This might be due to the small number of cells involved per experiment which prevents performing robust clustering.”

Remark 5:

* Commonly a scatter plot of R^2 is shown across models, where each cell is a point, rather than reporting the mean error alone. Could you scatter plot R^2 of LN vs CNN vs contrast model? This will give the reader the sense of the number of cells better explained vs if it's a few outlier cells (this seems unlikely but would be useful to show).

Authors: Here it is the scatter plot of the LN R^2 vs the CNN R^2 for all the modelled experiments:

Supplementary Figure 4. Scatter plot of model performance for all cells, in reference to Fig.2

The model performance of the CNN was higher than the LN performance in most of the cells that we model, both in mouse (blue) and axolotl (red).

The CNN outperforms the LN for the majority of the cells. Here it is the R^2 of the contrast model vs the R^2 of the CNN

Supplementary Figure 6. Scatter plot of CNN vs contrast model performance, in reference to Fig. 4.

The model performance of the CNN was higher than the contrast model performance in most of the cells that we model, both in mouse (blue) and axolotl (red), even though the contrast model captures the qualitative behavior of these cells (Fig. 4).

the cells are fewer because the contrast model makes sense only on the cells that show polarity inversion. The CNN outperforms the contrast for almost every cell.

Changes:

i) We added these scatter plots in the Supplementary Figure 4 and 6

ii) We referenced the reader to the supplementary figure in the caption of figure 2B: "(see Methods and Supplementary Fig. 4 for a scatter plot of this data)", and in the caption of figure 4G: "(see Supplementary Fig. 6 for a scatter plot of this data)".

REVIEWERS' COMMENTS

Reviewer #1 (Remarks to the Author):

The authors have thoroughly addressed my comments and questions from the previous round of reviews. This is an excellent manuscript and study that significantly advances our understanding of neural computations performed in retina.

Reviewer #2 (Remarks to the Author):

The authors thoroughly addressed all of my comments, and even added new experiments with non-static images that substantially improved the relevance of the work. This paper contributes new insights into the understanding of retinal computations.